# EDITOR: EFFECTIVE AND INTERPRETABLE PROMPT INVERSION FOR TEXT-TO-IMAGE DIFFUSION MODELS

## ABSTRACT

Text-to-image generation models (e.g., Stable Diffusion) have achieved significant advancements, enabling the creation of high-quality and realistic images based on textual descriptions. Prompt inversion, the task of identifying the textual prompt used to generate a specific artifact, holds significant potential for applications including data attribution, model provenance, and watermarking validation. Recent studies introduced a delayed projection scheme to optimize for prompts representative of the vocabulary space, though challenges in semantic fluency and efficiency remain. Advanced image captioning models or visual language models can generate highly interpretable prompts, but they often lack in image similarity. In this paper, we propose a prompt inversion technique called EDITOR for text-to-image diffusion models, which includes initializing embeddings using a pre-trained image captioning model, refining them through reverse-engineering in the latent space, and converting them to texts using an embedding-to-text model. Our experiments on the widely-used datasets, such as MS COCO, LAION, and Flickr, show that our method outperforms existing methods in terms of image similarity, textual alignment, prompt interpretability and generalizability. We further illustrate the application of our generated prompts in tasks such as cross-concept image synthesis, concept manipulation, evolutionary multi-concept generation and unsupervised segmentation. Code: `https://anonymous.4open.science/r/EDITOR`.

## 1 INTRODUCTION

Text-to-image generation models have made remarkable progress, greatly enhancing the creation of visual content. Advanced deep learning techniques enable models like *Parti* (Yu et al., 2022), *DALL-E* (Ramesh et al., 2022), and *Stable Diffusion* (Rombach et al., 2022) to produce high-quality, realistic, and creative images based on textual descriptions. A key factor in successfully generating high-quality images is the design of prompts. Well-crafted prompts guide these models to generate images that match the desired details and styles. As such, prompts used to generate high-quality images have become a critical component of the ecosystem and valuable IP assets (Wang et al., 2024a; Zhang et al., 2023; Shen et al., 2024).

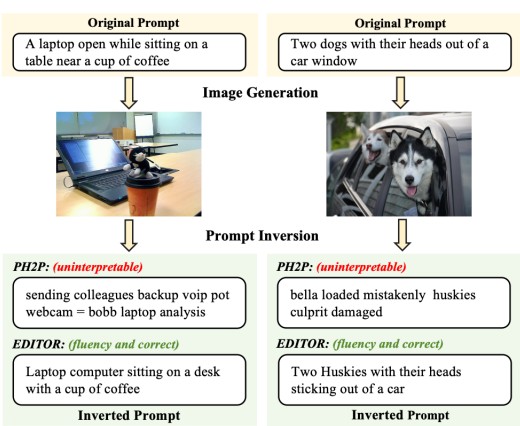

Figure 1: Original prompts and generated images along with inverted prompts.

Prompt inversion is the task of reconstructing the textual prompt used to generate a specific image. It has various applications in trustworthy artificial intelligence (AI), such as data attribution, model provenance, and watermarking validation (Wang et al., 2024b;c; 2023a; Naseh et al., 2024; Shen et al., 2024). However, as summarized in Table 1, current methods face **two main challenges**: they either lack *image similarity* or fall short in *prompt interpretability*.

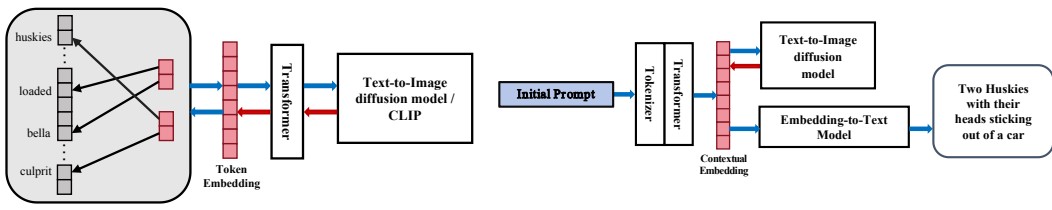

(a) Optimization of embedding in PEZ and PH2P.     (b) Optimization of embedding in EDITOR.

Figure 2: Existing works optimize the token embedding before transformer layer and project them to nearest token embeddings, often producing incoherent results; EDITOR optimizes the contextual embedding after transformer layer and converts the final optimized embeddings into prompts. The red arrow indicates the backward pass in gradient-based optimization.

A straightforward way to obtain interpretable prompts is to apply captioning models or advanced vision language models, such as BLIP-2 (Li et al., 2023), LLaVA-1.5 (Liu et al., 2024), or Janus-Pro (Chen et al., 2025). While these mod-

Table 1: Challenges of current prompt inversion.

| Methods | PEZ | PH2P | Image Captioning | EDITOR |
|---|---|---|---|---|
| Prompt Interpretability? | ✗ | ✗ | ✓ | ✓ |
| Image Similarity? | ✓ | ✓ | ✗ | ✓ |

els generate fluent descriptions, they fail to ensure high similarity when re-generating the image with a diffusion model. On the other hand, optimization-based approaches such as PEZ (Wen et al., 2024) and PH2P (Mahajan et al., 2024) update latent embeddings and repeatedly project them onto the vocabulary. However, these **discrete projections disrupt semantic continuity**, causing unreadable prompts, computational inefficiency. More importantly, the projection step incurs **severe embedding discrepancy**, with empirical results indicating that the cosine similarity between the optimized embedding and its projected counterpart often falls to only **0.167**. Such a drastic degradation implies that even if optimization successfully approaches a near-optimal embedding, *the subsequent projection may push the representation far away from this solution, deviating negatively from the optimum*. Consequently, the optimization process becomes unstable and less efficient, limiting the effectiveness of these methods in recovering high-quality prompts.

In this paper, we propose a new prompt inversion method, called EDITOR, for text-to-image diffusion models to effectively generate prompts that are both accurate and elegant. Instead of projecting the embeddings back to the prompts at every iteration as done by existing approaches (Wen et al., 2024; Mahajan et al., 2024), EDITOR directly **optimizes the representation in the continuous space** (e.g., contextual embeddings). This ensures that the optimized embeddings are optimal and can induce the intended image. To obtain discrete prompts, EDITOR leverages an embedding-to-text model (Morris et al., 2023) for projection. Unlike prior work, which simply projects embeddings to their nearest prompts—causing substantial discrepancies—we utilize text-representation pairs collected from the subject diffusion model to train our embedding-to-text model. Additionally, EDITOR is equipped with a correction model that iteratively refines prompts, pulling their embeddings closer to their originally optimized counterparts. This design ensures that *the optimized representation can be projected into an in-distribution space* and is therefore more semantically aligned with the original prompt. Figure 2 illustrates the comparison between existing approaches and our method EDITOR in optimizing embeddings and obtaining prompts. Figure 1 presents examples of our prompt inversion result and comparison with PH2P (Mahajan et al., 2024). Compared to the baselines, our method produces much more coherent and interpretable prompts. Furthermore, the reverse-engineering step ensures that the generated prompt remains semantically aligned with the target image, offering improved image quality compared to directly using an image captioning model.

Our results on the widely-used datasets (e.g., MS COCO (Lin et al., 2014), LAION (Schuhmann et al., 2022), Flickr (Young et al., 2014) and DiffusionDB (Wang et al., 2023b)) under state-of-the-art text-to-image generation model (e.g., Stable Diffusion (Rombach et al., 2022)) show that EDITOR is more effective in both image similarity and prompt quality than existing methods. Our contributions are summarized as follows:

- We introduce a novel technique called EDITOR for inverting prompts in text-to-image diffusion models. Unlike existing methods that rely on discrete optimization with token embedding projection onto the vocabulary—a process that disrupts semantic continuity and causes severe

embedding discrepancy—EDITOR improves optimization effectiveness and efficiency through contextual embedding optimization in a continuous latent space.

- EDITOR employs a novel three-step prompt inversion pipeline including initialization, revere-engineering and embedding inversion. This framwork surpasses existing prompt inversion methods like PEZ (Wen et al., 2024) , PH2P (Mahajan et al., 2024), STEPS (Qiu et al., 2025) and VGD (Kim et al., 2025) in prompt interpretability and outperforms advanced image captioning models and vision language models in image similarity.

- We demonstrate that EDITOR produces semantically meaningful prompts excelling in image similarity, text alignment, and interpretability across four standard datasets. In addition, EDITOR also achieves strong performance when applied to advanced multi-encoder diffusion models such as SDXL-Turbo and Stable Diffusion 3.5 Medium, showing its robustness across different archi-tectures. Moreover, they enable flexible downstream tasks such as cross-concept image synthesis, concept manipulation, evolutionary multi-concept generation and unsupervised segmentation.

## 2 RELATED WORK

**Text-to-Image Generation.** Early approaches to text-to-image generation were dominated by generative adversarial networks (GANs) (Zhang et al., 2017; Karras, 2019) to generate style-controllable images. Inspired by the success of autoregressive transformer (Vaswani, 2017), recent methods (Ding et al., 2021; Esser et al., 2021) generate images sequentially by modeling the conditional probability of pixels or patches. Significant progress in high-resolution image synthesis has been made with diffusion models (Ho et al., 2020) that gradually refines an initial noisy input into a coherent image. Conditional diffusion models (Ramesh et al., 2022; Saharia et al., 2022) leverage text encoders (Radford et al., 2021b; Raffel et al., 2020b) to map texts into latent embeddings for controllable generation. To improve efficiency, latent diffusion models (LDMs) (Rombach et al., 2022; Gu et al., 2022; Luo et al., 2023), perform the diffusion process in a compressed latent space instead of the pixel space. Given an image $x$, an encoder $\mathcal{E}(\cdot)$ compresses the image into a latent representation $z$, and a decoder $\mathcal{D}(\cdot)$ reconstructs the image back into pixel space, such that $x \approx \mathcal{D}(z) = \mathcal{D}(\mathcal{E}(x))$. Here, $z$ represents the latent representations of the image. During the diffusion process, noise is incrementally added to $z$, resulting in a sequence of latent representations $z_t$ for each timestep $t \in \{0, 1, \ldots, T\}$, where $T$ is the total number of timesteps. A U-Net (Ronneberger et al., 2015) $\epsilon_\theta(z_t, t)$ is trained as a denoising module to predict the noise component within $z_t$. In conditional text-to-image diffusion models, the denoising process is guided by conditional text inputs. A transformer-based text encoder $\mathcal{T}(\cdot)$ is employed to convert a given prompt $p$ into a corresponding latent text embedding $c := \mathcal{T}(p)$. The conditional U-Net $\epsilon_\theta$ takes $(z_t, t, c)$ as input, and the training objective of the conditional latent diffusion model is formalized as

$$\mathbb{E}_{z,c,\epsilon,t}\big[\|\epsilon_\theta(z_t, t, c) - \epsilon\|_2^2\big]. \tag{1}$$

with $\epsilon_\theta$ predicting the noise and $\epsilon$ denoting the ground-truth noise. During inference in the conditional diffusion model, we initialize the process with a latent seed, which is used to generate random latent noise $n$ of the same shape as $z_t$, serving as the starting point for denoising. The process of synthesizing an image can be expressed as $x = \mathcal{D}(\mathcal{R}_{\epsilon_\theta}(c, n)) = \mathcal{D}(\mathcal{R}_{\epsilon_\theta}(\mathcal{T}(p), n))$. where $\mathcal{R}_{\epsilon_\theta}$ is the generation process with $\epsilon_\theta$ in the latent space.

**Soft Prompt Inversion for Diffusion Models.** Previous inversion techniques in diffusion models aim to enhance controllability and personalization in text-to-image generation by embedding visual concepts into the model's textual or latent space (Gal et al., 2022; Voynov et al., 2023; Ruiz et al., 2023; Wei et al., 2023; Gal et al., 2023). DreamBooth (Ruiz et al., 2023) fine-tunes diffusion models using specific reference images to learn new visual features for personalized image generation. Textual inversion (Gal et al., 2022) encodes visual concepts as unique tokens in the model's vocabulary, enabling customized image generation through textual prompts. Gal et al. (2023) presents an encoder-based approach for fast personalization of text-to-image models. Similarly, ELITE (Wei et al., 2023) extends textual inversion by encoding multiple visual concepts into the textual embedding space, allowing the combination of various concepts within a single prompt. Moreover, Voynov et al. (2023) extends prompt conditioning, where different embeddings are injected into different layers of the U-net. Although these methods enhance personalization and controllability in text-to-image generation, they often lack interpretability and generalization, limiting their applicability to diverse contexts.

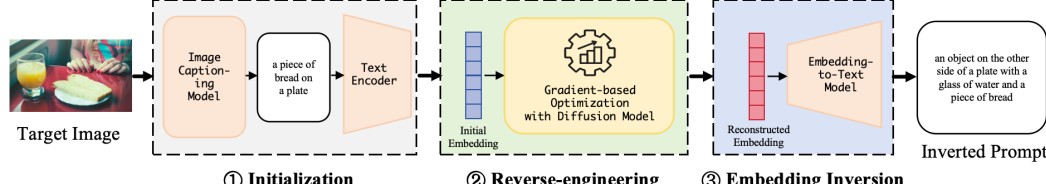

Figure 3: Overview of EDITOR. Our approach comprises three steps: ① initializing the latent embedding via a pre-trained image captioning model; ② refining the embedding through reverse-engineering; ③ mapping the refined embedding to text with an embedding-to-text model. This pipeline yields coherent prompts and streamlines optimization.

**Hard Prompt Inversion for Diffusion Models.** One solution to obtain the hard prompt is to invert image features into natural language representations. Inspired by discrete optimization, recent works (Wen et al., 2024; Mahajan et al., 2024; Qiu et al., 2025) focus on optimizing textual prompts to better align with image features. Wen et al. (2024) introduce a gradient-based discrete optimization method for hard prompt discovery with CLIP; Mahajan et al. (2024) propose hard prompt optimization that aligns prompts with image features directly in diffusion models; and STEPS (Qiu et al., 2025) formulates prompt search as a sequential probability tensor estimation problem that optimizes hard prompts under a fixed query budget. However, these inverted prompts often lack semantic fluency and naturalness, resulting in outputs unintelligible to humans. Another recent work, Visually Guided Decoding (VGD) (Kim et al., 2025), is gradient-free and uses a large language model plus CLIP-based guidance to generate coherent, human-readable prompts; it can generate fluent prompts efficiently but still falls short in terms of achieving high image similarity. In contrast, our method ensures high image similarity while making prompts more human-understandable, thereby enhancing their interpretability and generalizability.

## 3 METHODOLOGY

In this section, we present the method developed for prompt inversion in text-to-image diffusion models. We begin by outlining the problem formulation for the problem under study, followed by an introduction to our prompt inversion method.

**Problem Formulation.** We aim to invert prompts from text-to-image generation models. Following existing works (Wen et al., 2024; Mahajan et al., 2024), we formulate the problem as follows:

- **Engineer's Goal.** The goal of prompt inversion for a given image $x$ is to find the prompt $p^*$ that generates an image $\mathcal{M}(p^*)$ as close as possible to $x$. Formally, the objective can be expressed as constructing an inversion algorithm $\mathcal{A} : (\mathcal{M}, x) \mapsto P$, where $P$ represents the text space.
- **Engineer's Capability.** Same as Mahajan et al. (2024), we assume that we have white-box access to a latent diffusion model $\mathcal{M}$, allowing them to access intermediate outputs and compute the model's gradients. In addition, the engineer has access to other public models and tools, which aids in more efficient prompt optimization.

**Overview.** The overview of EDITOR is shown in Figure 3, consisting of three main steps: initialization, reverse-engineering, and embedding inversion. The overall gradient-based optimization algorithm is shown in Algorithm 1.

**Initialization for Latent Text Embedding.** We leverage pre-trained image captioning models to generate an initial prompt for the target image. This approach ensures that the inverted prompt begins with a semantically meaningful and contextually relevant prompt, thereby reducing the search space and aligning the embedding more closely with the target distribution. Specifically, for a given image $x$, we utilize an image captioning model to generate an initial prompt $p_0$. This initial prompt is then encoded into a starting point for optimization using the text encoder $\mathcal{T}$, which maps the text into the latent space of the diffusion model.

It is worth noting that the initialization strategy is unique to our method, benefiting from the optimization in a continuous latent space. Unlike hard prompt optimization methods such as PEZ (Wen et al., 2024) and PH2P (Mahajan et al., 2024), our approach does not significantly disrupt the syntactic structure of the initial prompt. In contrast, as shown in Table 8, even with initialization, PEZ and PH2P produce token combinations with high perplexity.

**Algorithm 1** Prompt Inversion for Text-to-Image Diffusion Model

**Input:** Image Decoder $\mathcal{D}$, Generation Process $\mathcal{R}$, U-Net $\epsilon_\theta$, Noise $n$, Image Captioning Model $\mathcal{M}_c$, E2T Model $\mathcal{M}_t$, Target Image $x$.

**Output:** Inverted Prompt

1: **function** INVERSION($\mathcal{M}, x$)
2:     $\mathcal{T} = \mathcal{M}$.Text Encoder
3:     ▷ Initialization
4:     $p \leftarrow \mathcal{M}_c(x)$
5:     $c \leftarrow \mathcal{T}(p)$
6:     ▷ Reverse-engineering
7:     **for** $e \leq$ max_epoch **do**
8:         $loss = \mathcal{L}\left(\mathcal{D}(\mathcal{R}_{\epsilon_\theta}(c, n)), x\right)$
9:         $\nabla_c = \frac{\partial loss}{\partial c}$
10:       $c = c - lr \cdot \nabla_c$
11:     **end for**
12:     ▷ Embedding Inversion
13:     **return** $\mathcal{M}_t(c)$
14: **end function**

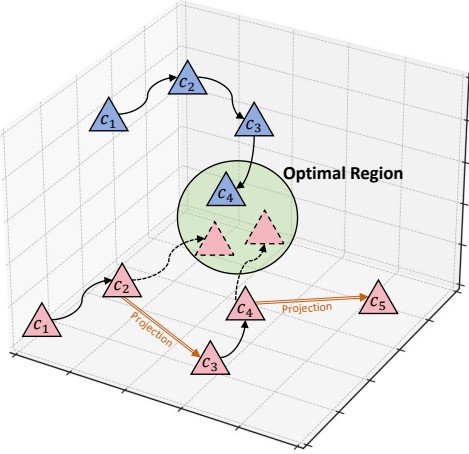

Figure 4: Comparison of optimization paths. The blue path illustrates **continuous optimization**. The red path shows **optimization with projection** in PEZ and PH2P, where each step is displaced due to vocabulary projection.

**Reverse-engineering for Latent Text Embedding.** With the initial prompt, we reconstruct latent text embeddings using input reverse-engineering. Given a target image $x$ and a latent diffusion model, the objective is to find an embedding $c^*$ such that the generated image $\mathcal{D}(\mathcal{R}_{\epsilon_\theta}(c^*, n))$ minimizes the distance to $x$. This input reverse-engineering task is formulated as:

$$c^* = \arg\min_c \mathcal{L}(\mathcal{D}(\mathcal{R}_{\epsilon_\theta}(c, n)), x). \tag{2}$$

where $\mathcal{L}$ denotes the mean squared error (MSE) loss, which computes the average squared difference between the generated and target images. The latent random noise $n$ is fixed, and the parameters of the latent diffusion model are held constant.

We use gradient-based optimization to iteratively search for the optimal embedding $c^*$ by updating the embedding based on the gradient of the reconstruction loss. For each step, the embedding $c$ is updated as follows:

$$c = c - lr \cdot \frac{\partial \mathcal{L}(\mathcal{D}(\mathcal{R}_{\epsilon_\theta}(c, n)), x)}{\partial c}. \tag{3}$$

where $lr$ is the learning rate. This process continues until the maximum number of epochs is reached.

In contrast to existing methods (Wen et al., 2024; Mahajan et al., 2024) that optimize the token embeddings before passing through the transformer layers and then project them onto the corresponding vocabulary, we directly optimize the output of the text encoder in the latent space. This approach offers two key advantages: (1) it enables more stable convergence by eliminating the repeated projection step. As illustrated in Figure 4, projection often displaces embeddings away from the optimal region, forcing the optimization to restart from a less favorable position. By avoiding this detour, EDITOR maintains the optimization trajectory within the continuous latent space, leading to significantly improved convergence speed and efficiency. (2) by optimizing the text encoder's output, which has already undergone deeper semantic processing, the generated text remains more natural, contextually relevant, and grammatically correct, while also avoiding the disjointed or redundant word choices that can arise in discrete optimization.

**Embedding Inversion.** It is also challenging to get text prompts from the reconstructed embedding, as the embedding is continuous and highly contextualized, making it impossible to be projected to the nearest token in the vocabulary, as is done in PEZ (Wen et al., 2024) and PH2P (Mahajan et al., 2024). To solve this problem, we leverage an embedding-to-text (E2T) model (Morris et al., 2023) to convert the reconstructed embedding back into text prompts. Let $c \in \mathbb{R}^d$ be a target (latent) embedding. We first train a *zero-step model* $M_{\text{zero}}$ that directly maps $c$ to text $p$, estimating the conditional probability $P_{\text{zero}}(p \mid c)$. This training uses a maximum likelihood objective:

$$\mathcal{L}_{\text{zero}} = -\sum_{(p,c)} \log P_{\text{zero}}(p \mid c). \tag{4}$$

Table 3: Image Similarity, Textual Alignment (BERTScore), and Prompt Interpretability comparison across all datasets, including DiffusionDB and the STEPS baseline.

| Dataset | Method | Image Similarity | | Textual Alignment | | | Prompt Interpretability |
|---|---|---|---|---|---|---|---|
| | | CLIP↑ | LPIPS↓ | Precision↑ | Recall↑ | F1↑ | PPL↓ |
| MS COCO | PEZ | 0.745 | 0.477 | 0.791 | 0.853 | 0.819 | 8,837.797 |
| | PH2P | 0.789 | 0.467 | 0.801 | 0.851 | 0.823 | 11,078.994 |
| | VGD | 0.725 | 0.509 | 0.826 | 0.885 | 0.853 | 665.418 |
| | STEPS | 0.687 | 0.480 | 0.770 | 0.847 | 0.805 | 5,298.071 |
| | EDITOR (Ours) | **0.796** | **0.414** | **0.900** | **0.920** | **0.908** | **80.659** |
| LAION | PEZ | 0.756 | 0.455 | 0.783 | 0.812 | 0.797 | 8,040.401 |
| | PH2P | **0.826** | 0.427 | 0.794 | 0.810 | 0.802 | 9,621.671 |
| | VGD | 0.769 | 0.466 | 0.800 | 0.818 | 0.808 | 700.754 |
| | STEPS | 0.722 | 0.476 | 0.769 | 0.796 | 0.782 | 7,040.633 |
| | EDITOR (Ours) | **0.826** | **0.401** | **0.841** | **0.824** | **0.832** | **187.299** |
| Flickr | PEZ | 0.722 | 0.442 | 0.774 | 0.857 | 0.811 | 9,425.353 |
| | PH2P | 0.755 | 0.439 | 0.798 | 0.855 | 0.824 | 11,804.100 |
| | VGD | 0.715 | 0.479 | 0.826 | 0.885 | 0.852 | 536.712 |
| | STEPS | 0.684 | 0.466 | 0.771 | 0.851 | 0.807 | 6,279.372 |
| | EDITOR (Ours) | **0.776** | **0.424** | **0.892** | **0.915** | **0.900** | **90.320** |
| DiffusionDB | PEZ | 0.766 | 0.466 | 0.800 | 0.807 | 0.803 | 10,896.730 |
| | PH2P | 0.742 | 0.459 | 0.766 | 0.795 | 0.780 | 7,286.705 |
| | VGD | 0.780 | 0.454 | 0.806 | 0.814 | 0.810 | 553.287 |
| | STEPS | 0.749 | 0.461 | 0.777 | 0.807 | 0.792 | 7,616.533 |
| | EDITOR (Ours) | **0.807** | **0.385** | **0.835** | **0.813** | **0.823** | **105.185** |

Once trained, $M_{\text{zero}}$ generates an *initial hypothesis* $\hat{p}$ by sampling from $P_{\text{zero}}(p \mid c)$. Next, we introduce a *correction model* $M_{\text{corr}}$ to iteratively refine $\hat{p}$. In each refinement step $k$, the model produces a revised text $p^{(k)}$ conditioned on 1) the target embedding $c$, 2) the current hypothesis $p^{(k-1)}$, and 3) the embedding $\mathcal{T}(p^{(k-1)})$. Formally:

$$P_{\text{corr}}\big(p^{(k)} \mid c; p^{(k-1)}; \mathcal{T}(p^{(k-1)})\big). \tag{5}$$

Here $\mathcal{T}(\cdot)$ denotes the text encoder that maps text to the same latent space as $c$. The correction model is realized with a transformer-based encoder-decoder and is trained via maximum likelihood loss:

$$\mathcal{L}_{\text{corr}} = - \sum_{(p^{(k)}, c)} \log P_{\text{corr}}(p^{(k)} \mid c). \tag{6}$$

During inference, beam search is applied to generate multiple refined hypotheses, and the final output is selected to minimize the distance $\big\| \mathcal{T}(p) - c \big\|$ with respect to the ground-truth embedding $c$.

We train our embedding-to-text model using text-representation pairs generated by the text encoder of the diffusion model. This design ensures that the optimized representations are mapped back into an in-distribution space, making them more semantically aligned with the original prompts.

Compared to PEZ (Wen et al., 2024) and PH2P (Mahajan et al., 2024), our embedding-to-text process also achieves far lower embedding discrepancy. As shown in Table 2, projection onto the vocabulary space incurs severe embedding

Table 2: Vocabulary Projection vs. Embedding Inversion.

| Metric | Vocabulary Projection | Embedding Inversion |
|---|---|---|
| Cosine Similarity ↑ | 0.167 | **0.737** |
| L2 Distance ↓ | 32.934 | **11.373** |

discrepancy (cosine similarity of only 0.1671 and L2 distance of 32.93), whereas our embedding inversion achieves markedly better alignment (cosine similarity of 0.7370 and L2 distance of 11.37). This quantitative gap indicates that direct projection not only disrupts semantic continuity but also restricts the optimization process, preventing embeddings from approaching the optimal solution.

# 4 EXPERIMENT SETUP

**Datasets.** We evaluate our method on four widely-used datasets: MS COCO (Lin et al., 2014), which contains diverse natural images with detailed annotations; LAION (Schuhmann et al., 2022), a large-scale dataset of image–text pairs; Flickr30k (Young et al., 2014), which consists of diverse images including everyday life scenes, natural landscapes, animals, urban settings, and human activities; and DiffusionDB (Wang et al., 2023b), a large-scale gallery of real user prompts and generated images

from text-to-image diffusion models, covering a much broader range of prompt styles and visual content. For each dataset, we randomly sample 100 images for evaluation.

**Models.** We employ Stable Diffusion v1.5 (Rombach et al., 2022) as our target text-to-image generative model for prompt inversion experiments. In addition, we compare our method against different image captioning models and vision language models, including LLaVA-1.5-7B (Liu et al., 2024), Janus-Pro-7B (Chen et al., 2025), BLIP-Large (Li et al., 2022b), BLIP2-OPT-2.7B (Li et al., 2023), and GIT-Large (Wang et al., 2022), and further conduct experiments on advanced multi-encoder architectures including SDXL-Turbo (Sauer et al., 2023) and Stable Diffusion 3.5 Medium (Esser et al., 2024). More detailed descriptions of these models can be found in the Appendix A.

**Evaluation Metrics.** We employ multiple metrics to comprehensively evaluate the performance of prompt inversion:

- **Image Similarity**: We utilize CLIP (Radford et al., 2021a) and LPIPS (Zhang et al., 2018) scores to measure the visual similarity between generated and target images, assessing the reconstruction quality.
- **Textual Alignment**: BERTScore (Zhang et al., 2020) is used to evaluate semantic consistency between inverted and ground-truth prompts, computing Precision, Recall, and F1 metrics.
- **Prompt Interpretability**: We calculate Perplexity(PPL) scores to assess the linguistic coherence of inverted prompts, where lower scores indicate more comprehensible text.

**Baseline Methods.** We compare our approach with state-of-the-art prompt inversion methods: PEZ (Wen et al., 2024), which inverts prompts via image–text similarity matching; PH2P (Mahajan et al., 2024), which employs a delayed projection scheme to optimize prompts for text-to-image diffusion models; VGD (Kim et al., 2025), which uses a large language model plus CLIP-based guidance to generate coherent, human-readable prompts; and STEPS (Qiu et al., 2025), which formulates prompt search as a sequential probability tensor estimation problem.

## 5 EXPERIMENT RESULTS

**Image Similarity.** We evaluate the effectiveness of the inverted prompts generated by EDITOR for image generation. For a given target image, we first perform inversion to recover the corresponding prompt and assess the relevance of the images generated by the diffusion model using the recovered prompt. We compare EDITOR with PEZ, PH2P, VGD and STEPS under different datasets. The results are shown in Table 3. For CLIP score, EDITOR achieves the highest scores. In detail, the average CLIP similarity score under MSCOCO, LAION and Flickr are 0.794, 0.826 and 0.776, respectively, outperforming PEZ, PH2P, VGD and STEPS. Furthermore, EDITOR achieves a value of 0.414 under MSCOCO, 0.401 under LAION and 0.424 under Flickr in terms of the LPIPS score, consistently outperforming the baselines. The results demonstrate that EDITOR's prompts achieve high reconstruction quality in terms of similarity to the target images.

**Textual Alignment.** We evaluate the semantic similarity between the inverted prompts and the ground-truth prompts using BERTScore (Zhang et al., 2020). As shown in Table 3, EDITOR achieves consistently higher precision, recall, and F1 scores compared to the baseline methods across all datasets. Notably, under MS COCO, LAION and Flickr, EDITOR achieves a precision of 0.900, 0.841, and 0.892, surpassing the baselines by a value of 0.074, 0.041 and 0.066. When considering recall, EDITOR also excels across datasets. In detail, EDITOR can achieve a recall score of 0.920, 0.824 and 0.915 under all four datasets, respectively, outperforming the baselines. These results illustrate the effectiveness of our approach in achieving semantically meaningful text recovery that aligns closely with the ground-truth prompts.

**Prompt Interpretability.** We further assess the interpretability of the recovered prompts by computing their perplexity scores (PPL). Lower perplexity indicates that the inverted prompts are more fluent, natural, and interpretable. Results in Table 3 demonstrates that EDITOR shows 110% and 104% improvement compared to PEZ, PH2P and STEPS on MS COCO and Flickr. Moreover, even when compared with VGD, which leverages large language models to generate fluent prompts with relatively low perplexity (665.418 under MS COCO, 700.754 under LAION, and 536.712 under Flickr), our method still achieves much lower scores. The vast gap in performance highlights the

Table 4: Images generated with inverted prompts. For a given target image (left), inverted prompts from PEZ Wen et al. (2024), PH2P Mahajan et al. (2024) and our method are used to prompt Stable Diffusion v1.5 Rombach et al. (2022), DALL-E 3 Betker et al. (2023) and Ideogram 2.0 (right).

| Target Image | Generated Images | | | Target Image | Generated Images | | |
|---|---|---|---|---|---|---|---|
| | Stable Diffusion | Dall-E 3 | Ideogram 2.0 | | Stable Diffusion | Dall-E 3 | Ideogram 2.0 |

**PEZ**: beach firing fortnight !), lgbti takeaways deploy acquisition labs recognised xerirrigation

**PH2P**: petersburg adid micropoetry beach houses shack houses spend out sunbathing living foodand

**EDITOR**: Affraies sitting in front of gloomy and colorful shacks on a beach

**PEZ**: hiber❉ unforgettable ophthalstowe maine mansion during giftideas indulge travelunited-states

**PH2P**: ride mice liking dorchester russian mansion attemp population ? yann reduced exception

**EDITOR**: Afraed House with a lit porch and balcony in the snow

**PEZ**: contemplating forests reignhormone blame millennials cleanenergy !), scouts immunology scientists

**PH2P**: outdoors hamp resignation sometimes backyard smokey dreaming relationships forest forest mountain exploring

**EDITOR**: a woman in a red dress standing on a fallen tree in the woods

**PEZ**: ajes pocheña ados psv skull pave gold ceramic watch mal ados

**PH2P**: tattoo watches sugar skull skull true housewives grand ( ese patriarch

**EDITOR**: a sugar skull watch in a pink box

Table 5: Comparison with advanced image captioning models or vision language models.

| Metric | EDITOR | LLaVA-1.5-7B | Janus-Pro-7B | BLIP-Large | BLIP2-OPT-2.7B | GIT-Large |
|---|---|---|---|---|---|---|
| CLIP Score↑ | **0.776** | 0.731 | 0.742 | 0.720 | 0.691 | 0.681 |
| LPIPS Score↓ | **0.424** | 0.464 | 0.460 | 0.451 | 0.496 | 0.504 |

exceptional ability of EDITOR to recover prompts that are not only semantically aligned with the ground-truth but also significantly more fluent and interpretable.

**Comparison with Image Captioning Models.** To demonstrate the superior performance of EDITOR compared to state-of-the-art image captioning models or vision language models, we evaluated their differences in image generation quality. As shown in Table 5, EDITOR achieves the highest CLIP Similarity score (0.776) and the lowest LPIPS Similarity score (0.424) when compared to other image captioning models. This indicates that the prompts generated by EDITOR have a stronger connection to the visual content of the images and are more consistent with human-labeled captions.

**Performance on Multi-Encoder Diffusion Models.** In addition to single-encoder diffusion models, we also evaluate EDITOR on advanced multi-encoder models, which integrate multiple text encoders to enhance text-image alignment. As shown in Table 6, EDITOR consistently achieves strong results across different model architectures. For instance, EDITOR obtains a CLIP score of 0.792 on SDXL-Turbo and 0.785 on Stable Diffusion 3.5 Medium, both close to the performance of the single-encoder Stable Diffusion v1.5 . Moreover, the generated prompts maintain low perplexity,

Table 6: Evaluation on advanced multi-encoder models.

| | CLIP Score ↑ | LPIPS Score ↓ | Prompt Precision ↑ | Prompt Recall ↑ | Prompt F1 ↑ | Prompt PPL ↓ |
|---|---|---|---|---|---|---|
| Stable Diffusion v1.5 | 0.829 | 0.426 | 0.842 | 0.825 | 0.834 | 154.362 |
| SDXL Turbo | 0.792 | 0.428 | 0.824 | 0.818 | 0.821 | 105.114 |
| Stable Diffusion 3.5 Medium | 0.785 | 0.445 | 0.814 | 0.817 | 0.815 | 115.132 |

Table 7: Impact of initialization.

| Metric | EDITOR w/o initialization | EDITOR |
|---|---|---|
| CLIP Score↑ | 0.658 | 0.784 |
| LPIPS Score↓ | 0.476 | 0.434 |
| Prompt Precision↑ | 0.815 | 0.863 |
| Prompt Recall↑ | 0.843 | 0.886 |
| Prompt F1↑ | 0.829 | 0.873 |
| Prompt PPL↓ | 158.770 | 103.485 |

Table 8: Results of PEZ and PH2P with initialization.

| Metric | PEZ | PEZ+Initialization | PH2P | PH2P+Initialization |
|---|---|---|---|---|
| CLIP Score↑ | 0.714 | 0.713 | 0.735 | 0.744 |
| LPIPS Score↓ | 0.456 | 0.426 | 0.432 | 0.460 |
| Prompt Precision↑ | 0.790 | 0.851 | 0.795 | 0.793 |
| Prompt Recall↑ | 0.857 | 0.889 | 0.849 | 0.850 |
| Prompt F1↑ | 0.820 | 0.867 | 0.819 | 0.818 |
| Prompt PPL↓ | 15,662.859 | 2,953.184 | 9,164.067 | 7,350.016 |

indicating that the outputs remain interpretable and linguistically coherent even under more complex architectures. These results suggest that EDITOR is not limited to a specific model design and can generalize well across both single-encoder and multi-encoder diffusion models, highlighting its robustness and broad applicability.

**Qualitative Results.** We show example inverted prompts towards four target images and corresponding generations with three models: Stable Diffusion (Rombach et al., 2022), DALL-E 3 (Betker et al., 2023), and Ideogram 2.0 in Table 4. EDITOR produces more accurate, readable prompts that generate images closely aligned with the target images, demonstrating strong generalizability across models. In contrast, prompts from PEZ and PH2P tend to be less coherent, often resulting in images that do not align well with the intended concepts, highlighting the superior precision and versatility of our approach.

## 6 ABLATION STUDIES

**Impact of Initialization.** To show the importance of initialization in EDITOR, we compared random initialization and image captioning model initialization on a Flickr subset. As shown in Table 7, random prompts hinder convergence, often leading to poor performance and local optima traps. Our initialization is crucial for guiding the algorithm to accurate, optimal solutions. For fairness, we also initialized PEZ and PH2P. As shown in Table 8, initialization only slightly improved PEZ's textual Alignment and prompt interpretability, with almost no impact on other aspects for both methods. This is because PEZ and PH2P are discrete optimization algorithms; even if they start with interpretable prompts, the optimization process may gradually replace them with high-perplexity token combinations. This also demonstrates that our method has greater potential than existing baselines, particularly in its compatibility with initialization.

**Impact of Correction Model in Embedding Inversion.** We further analyze the role of the correction Model $M_{\text{corr}}$ in our embedding inversion stage. While the Zero-step Model $M_{\text{zero}}$ already produces a coherent initial prompt by directly decoding the optimized contextual embedding, this one-shot decoding can still leave minor semantic drift relative to the target embedding. Incorporating the correction model substantially mitigates this issue. As shown in Table 9, adding $M_{\text{corr}}$ consistently improves CLIP similarity, F1, and LPIPS across all four datasets (MS COCO, LAION, Flickr, and DiffusionDB). The correction stage iteratively refines the decoded prompt so that its re-encoded embedding more closely matches the optimized embedding, effectively reducing residual embedding discrepancy. These consistent improvements demonstrate that the correction Model $M_{\text{corr}}$ is essential for achieving the highest reconstruction fidelity and textual accuracy.

## 7 APPLICATION OF THE GENERATED PROMPTS

**Cross-Concept Image Synthesis.** Our method combines multiple prompts into a unified prompt, enabling seamless image fusion. This supports applications like composite image generation and multi-concept scene creation, preserving semantic and visual coherence. For example, Figure 5 shows the fusion of two prompts: *"a woman in a red dress standing on a fallen tree in the woods"* and *"colorful shacks on a beach"*. The resulting image integrates the woman into a beach shack scene, maintaining prompt accuracy and visual integrity.

Table 9: Impact of Correction model in Embedding Inversion.

| Dataset | Model | CLIP ↑ | LPIPS ↓ | Precision ↑ | Recall ↑ | F1 ↑ | PPL ↓ |
|---|---|---|---|---|---|---|---|
| MS COCO | Zero Step Model | 0.772 | 0.414 | 0.898 | 0.910 | 0.903 | 105.906 |
| | Zero Step + Correction Model | 0.796 | 0.414 | 0.900 | 0.920 | 0.908 | 80.659 |
| LAION | Zero Step Model | 0.811 | 0.391 | 0.832 | 0.820 | 0.826 | 214.636 |
| | Zero Step + Correction Model | 0.826 | 0.401 | 0.841 | 0.824 | 0.832 | 187.299 |
| Flickr | Zero Step Model | 0.764 | 0.402 | 0.885 | 0.913 | 0.896 | 100.287 |
| | Zero Step + Correction Model | 0.776 | 0.424 | 0.892 | 0.915 | 0.900 | 90.320 |
| DiffusionDB | Zero Step Model | 0.803 | 0.412 | 0.829 | 0.817 | 0.822 | 110.294 |
| | Zero Step + Correction Model | 0.807 | 0.385 | 0.835 | 0.813 | 0.823 | 105.185 |

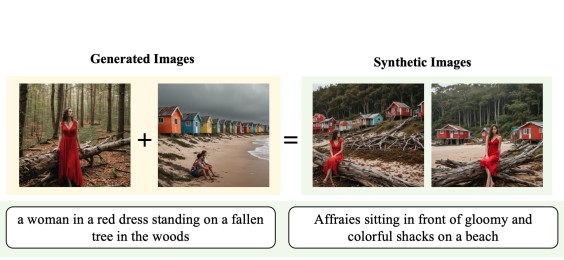

Figure 5: Application of cross-concept image synthesis.

Figure 6: Application of removal or replacement of objects.

**Removal or Replacement of Concepts.** Our prompt inversion method enables removal or replacement of image concepts. Unlike PEZ (Wen et al., 2024) and PH2P (Mahajan et al., 2024), which produce hard-to-interpret prompts, our method generates clear, human-readable prompts, simplifying token identification for concept editing. This improves flexibility in image manipulation. For example, Figure 6 shows removing "trees" from the prompt eliminates them, while replacing "trees" with "fence" substitutes a fence, demonstrating effective prompt-based editing. For more applications including evolutionary multi-concept generation and unsupervised segmentation, refer to Appendix D.

## 8    CONCLUSION

In this paper, we introduced a prompt inversion approach for text-to-image diffusion models that generates high-quality, semantically aligned and interpretable prompts. Our method, validated on datasets like MS COCO, LAION, Flickr and DiffusionDB, outperforms existing techniques in image similarity, textual alignment, and prompt interpretability. Additionally, the generated prompts are versatile and applicable to tasks such as cross-concept image synthesis and object removal or replacement. These results demonstrate that prompt inversion is a promising technique for improving the performance and versatility of text-to-image generation models.

## ETHICS STATEMENT

As the economic and creative value of prompts in text-to-image generation models grows, there are rising ethical concerns regarding prompt inversion techniques. Our method is capable of recovering high-quality prompts from generated images, which raises potential risks of prompt stealing. Such misuse could infringe upon the intellectual property of prompt engineers and undermine the commercial ecosystem of the prompt marketplace (Shen et al., 2024; Naseh et al., 2024). We acknowledge this concern and emphasize that our work is intended to advance the understanding of this emerging issue and to provide insights into potential safeguards for protecting prompt intellectual property, rather than to facilitate malicious exploitation.

In addition, our method can be applied to data attribution and model provenance. While these applications have positive implications for accountability and transparency, they also highlight broader ethical considerations about privacy, fairness, and legal compliance when tracing data origins. To mitigate risks, we encourage responsible use of EDITOR within controlled research and verification settings. We are committed to ensuring that this research contributes constructively to the dialogue on intellectual property protection, responsible AI, and ethical deployment of generative models.

## REPRODUCIBILITY STATEMENT

We have made significant efforts to ensure the reproducibility of our work. All details of our proposed method, including the initialization procedure, optimization strategy, and embedding-to-text (E2T) model design, are fully described in section 3 and Appendix C. Comprehensive experimental settings, including datasets, pevaluation metrics, implementation details and the model used in the experiments are documented in section 4 and Appendix A. In addition, ablation studies provided in Appendix E further clarify the role of each component of our pipeline. An anonymous link to the source code and trained models is included in the abstract to facilitate replication of our experiments.

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

# A    MORE DETAILS OF THE USED MODELS

In this section, we provide more details about the model used in the experiments.

We conduct experiments on three representative text-to-image generative models with different architectural designs, including Stable Diffusion v1.5(Rombach et al., 2022), SDXL Turbo(Sauer et al., 2023), and Stable Diffusion 3.5 Medium(Esser et al., 2024).

- **Stable Diffusion v1.5**[1]. This model is initialized with the weights of the Stable-Diffusion-v1.2[2] checkpoint and subsequently fine-tuned on 595k steps at resolution 512x512 on "laion-aesthetics v2 5+" and 10% dropping of the text-conditioning to improve classifier-free guidance sampling. The text encoder of this model is CLIP-ViT/L.
- **SDXL Turbo**[3]. This model is a distilled version of SDXL 1.0[4], trained for real-time synthesis. SDXL-Turbo is based on a novel training method called Adversarial Diffusion Distillation (ADD), which allows sampling large-scale foundational image diffusion models in 1 to 4 steps at high image quality. This approach uses score distillation to leverage large-scale off-the-shelf image diffusion models as a teacher signal and combines this with an adversarial loss to ensure high image fidelity even in the low-step regime of one or two sampling steps. The text encoders of this model are OpenCLIP-ViT/G and CLIP-ViT/L.
- **Stable Diffusion 3.5 Medium**[5]. This model is a Multimodal Diffusion Transformer with improvements (MMDiT-X) text-to-image model that features improved performance in image quality, typography, complex prompt understanding, and resource-efficiency. The text encoders of this model are OpenCLIP-ViT/G, CLIP-ViT/L and T5-xxl.

In addition to text-to-image diffusion models, we also compare our method with several representative image captioning models and vision-language models that are widely used for generating natural language descriptions from images, including LLaVA-1.5-7B (Liu et al., 2024), Janus-Pro-7B (Chen et al., 2025), BLIP-large (Li et al., 2022b), BLIP2-OPT-2.7B (Li et al., 2023) and GIT-large (Wang et al., 2022).

- **LLaVA-1.5-7B**[6]. This model is an open-source chatbot trained by fine-tuning LlamA/Vicuna on GPT-generated multimodal instruction-following data. It is an auto-regressive language model, based on the transformer architecture. In other words, it is an multi-modal version of LLMs fine-tuned for chat/instructions.
- **Janus-Pro-7B**[7]. This model is a novel autoregressive framework that unifies multimodal understanding and generation. It addresses the limitations of previous approaches by decoupling visual encoding into separate pathways, while still utilizing a single, unified transformer architecture for processing. The decoupling not only alleviates the conflict between the visual encoder's roles in understanding and generation, but also enhances the framework's flexibility. Janus-Pro surpasses previous unified model and matches or exceeds the performance of task-specific models. The simplicity, high flexibility, and effectiveness of Janus-Pro make it a strong candidate for next-generation unified multimodal models.
- **BLIP-Large**[8]. This model is a vision-language pretraining (VLP) framework designed for both understanding and generation tasks. Most existing pretrained models are only good at one or the other. It uses a captioner to generate captions and a filter to remove the noisy captions. This increases training data quality and more effectively uses the messy web data.
- **BLIP2-OPT-2.7B**[9]. This model is initialized with weights of the image encoder and large language model from pre-trained checkpoints and keep them frozen while training the Querying Transformer, which is a BERT-like Transformer encoder that maps a set of "query tokens" to query embeddings, which bridge the gap between the embedding space of the image encoder and

---

[1]https://huggingface.co/stable-diffusion-v1-5/stable-diffusion-v1-5

[2]https://huggingface.co/CompVis/stable-diffusion-v1-2

[3]https://huggingface.co/stabilityai/sdxl-turbo

[4]https://huggingface.co/stabilityai/stable-diffusion-xl-base-1.0

[5]https://huggingface.co/stabilityai/stable-diffusion-3.5-medium

[6]https://huggingface.co/llava-hf/llava-1.5-7b-hf

[7]https://huggingface.co/deepseek-ai/Janus-Pro-7B

[8]https://huggingface.co/Salesforce/blip-image-captioning-large

[9]https://huggingface.co/Salesforce/blip2-opt-2.7b

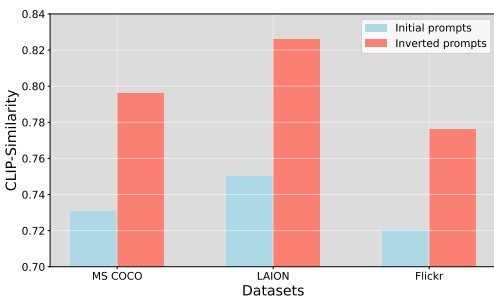 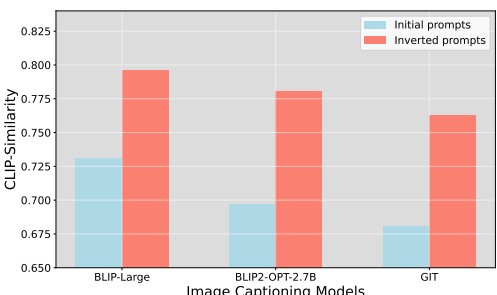

Figure 7: Comparison of CLIP-Similarity for initial and inverted prompts across Datasets.

Figure 8: Comparison of CLIP-Similarity for initial and inverted prompts across image captioning models.

the large language model. The goal for the model is simply to predict the next text token, giving the query embeddings and the previous text.

- **GIT-Large**[10]. This model is a transformer decoder conditioned on both CLIP image tokens and text tokens. The model is trained using "teacher forcing" on a lot of (image, text) pairs. The goal for the model is simply to predict the next text token, giving the image tokens and previous text tokens.

## B  IMPROVING IMAGE CAPTIONING WITH EDITOR

An important characteristic of EDITOR is its ability to optimize captions generated by an image captioning model in conjunction with a target diffusion model. This leads to improved text-image alignment. Figure 7 shows the enhancements achieved using EDITOR across datasets. The results clearly demonstrate that EDITOR significantly enhances the similarity score between generated images and the original images, increasing the CLIP similarity by at least 5%. Figure 8 shows the improvements achieved using EDITOR across different image captioning models. The results indicate that EDITOR can significantly improve the quality of generated images across various captioning models. What's more, we can fully believe that if image captioning models or vision-language models have nice performance, EDITOR can achieve better results based on them.

## C  DETAILS ABOUT THE E2T MODEL

Table 10: Effectiveness of the Embedding-to-Text (E2T) Model.

| Precision | Recall | F1 Score |
|---|---|---|
| 0.968 | 0.971 | 0.969 |

For the embedding-to-text (E2T) model, we use T5-base[11] (Raffel et al., 2020a) as the backbone, and the training was performed on the MSMARCO corpus (Bajaj et al., 2016), which contains 8.84M text–representation pairs, with a batch size of 32, a learning rate of $1 \times 10^{-3}$, a maximum token length of 32, an epoch number of 60 for the zero-step model and an epoch number of 35 for the correction model. The E2T model learns a general mapping from the diffusion model's embedding space to natural language, and does not introduce any additional information about the prompt inversion task. This training process is completely decoupled from the main prompt inversion task and its datasets.

We test the E2T model using 2,500 prompts selected from the MS COCO dataset (Lin et al., 2014), with evaluation metrics based on BertScore (Zhang et al., 2020), including Precision (P), Recall (R), and F1-score. The results was shown in Table 10. Our E2T model achieves over 0.968 in Precision, Recall, and F1-score, demonstrating that our model can accurately reverse the embedding back into the original prompt with high precision.

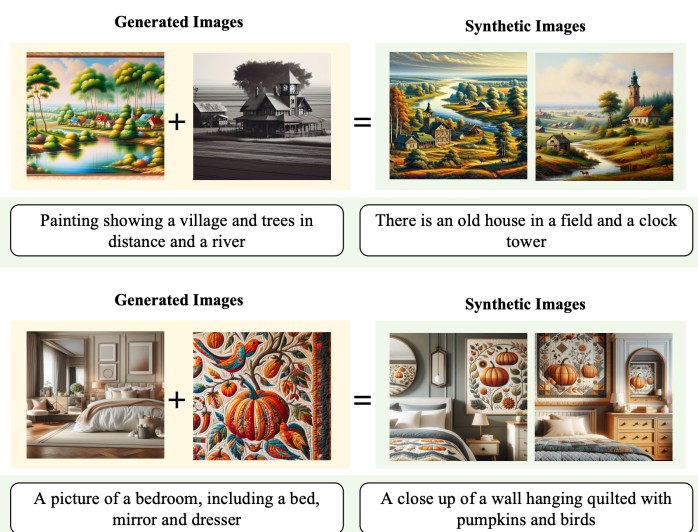

Figure 9: Application of cross-concept image synthesis.

# D    APPLICATIONS OF PROMPT INVERSION

**Cross-Concept Image Synthesis.** Our method enables the concatenation of multiple prompts into a single, unified prompt, allowing for the seamless fusion of corresponding images. This ability enhances creative possibilities, enabling applications such as composite image generation, multi-concept scene creation, and image synthesis from diverse ideas, all while maintaining semantic integrity and visual coherence across the fused images. For example, Figure 9 demonstrates the fusion of two distinct visual concepts using concatenated prompts. The first part of the prompt describes *"Painting showing a village and trees in distance and a river"* while the second part introduces *"There is an old house in a field and a clock tower"*. By combining these two prompts, our method generates a coherent synthetic image that seamlessly integrates the village landscape with the architectural elements of the old house and clock tower, while maintaining the visual style of both inputs. Similarly, the second example fuses *"A picture of a bedroom, including a bed, mirror and dresser"* with *"A close up of a wall hanging quilted with pumpkins and birds"*. The resulting composite image naturally incorporates the pumpkin-and-bird textile pattern into the bedroom scene, demonstrating that our approach can blend disparate concepts into a unified and semantically consistent output.

**Removal or Replacement of Concepts.** Our prompt inversion method can also be applied to the removal or replacement of concepts within an image. Unlike PEZ Wen et al. (2024) and PH2P Mahajan et al. (2024), which often generate prompts that are difficult to interpret, making it challenging to identify the corresponding tokens for the objects to be replaced, our method generates prompts that are fully human-readable and understandable. This clarity allows for easy identification of the specific tokens associated with the concepts, facilitating the seamless replacement or removal of those concepts within the image. This advantage enhances the flexibility and usability of our approach in image manipulation tasks. As shown in Figure 10, removing the word "egg" from the prompt eliminates them from the image, while replacing "egg" with "broccoli" substitutes the egg with a broccoli. This demonstrates the flexibility of our method in editing images via prompt modification.

**Unsupervised Segmentation.** EDITOR can be applied to unsupervised semantic segmentation by leveraging cross-attention maps to create segmentation masks Wu et al. (2023); Chefer et al. (2023). EDITOR generates prompts capturing key concepts directly from a diffusion model, without external data. For a target image, we use EDITOR to produce the inverted prompt, and as shown in Figure 11, the resulting cross-attention maps align tokens with relevant image regions, enabling accurate segmentation.

---

[10]https://huggingface.co/microsoft/git-large

[11]https://huggingface.co/google-t5/t5-base

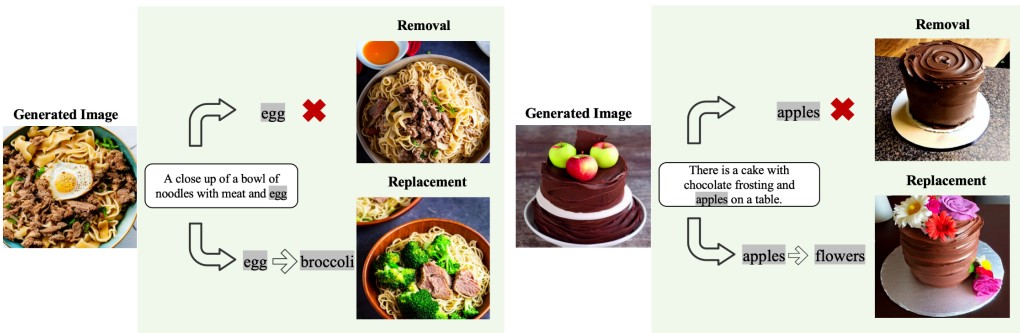

Figure 10: Application of removal or replacement of objects.

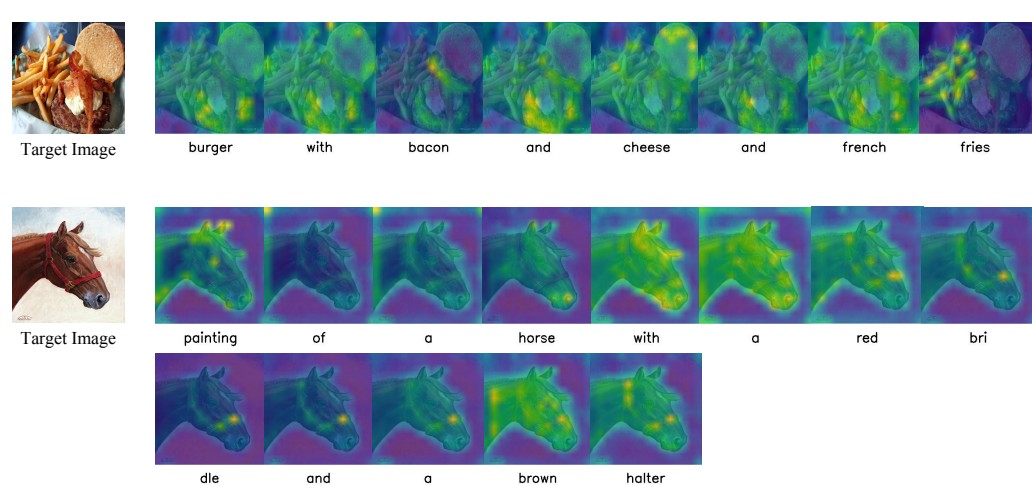

Figure 11: Application of unsupervised segmentation.

**Evolutionary Multi-concept Generation.** Evolutionary multi-concept image generation is a process that enables users to create complex images by iteratively combining concepts from multiple sample images using prompt inversion. Consider a example in Figure 12, we used EDITOR to get two prompts from two images: *"a giraffe burger with bacon and cheese and french fries"* and *"there is a bowl of spinach and some tomatoes in it."* We then combined these two prompts to create a new image, which resulted in a new prompt through EDITOR: *"Burger with fries on a plate with salad and tomatoes on it."*.

## E    ABLATION STUDIES

**Impact of Initial Prompt Length.** Since our prompt inversion method starts with an initial prompt, we explore the impact of the initial prompt length on the inversion performance. We vary the initial prompt length from 12 to 20 tokens, and the results are shown in Table 11. The results show that an initialization length of 16 tokens achieves the best performance. Shorter lengths (e.g., 12 tokens) tend to lack sufficient semantic information, making it harder to accurately reconstruct prompts. In contrast, longer prompts (e.g., 20 tokens) provide richer semantic information, leading to improved textual alignment and prompt interpretability. For instance, EDITOR achieves a prompt precision of 0.884 with 20 tokens, surpassing the precision of 0.852 achieves with 12 tokens. However, longer prompts introduce excessive complexity, making the optimization process more challenging. Compared with 20 tokens, the result of 16 tokens achieves a higher CLIP Similarity of 0.789 and better performance in textual alignment and prompt interpretability. At the same time, this length also offers a lower optimization complexity. As a result, an appropriate initialization length such as 16 tokens could be selected to balance semantic richness and optimization efficiency.

**Impact of the Number of Iteration.** We explore the impact of the number of iteration during the optimization for embedding inversion in Table 12. Generally, increasing the number of iterations

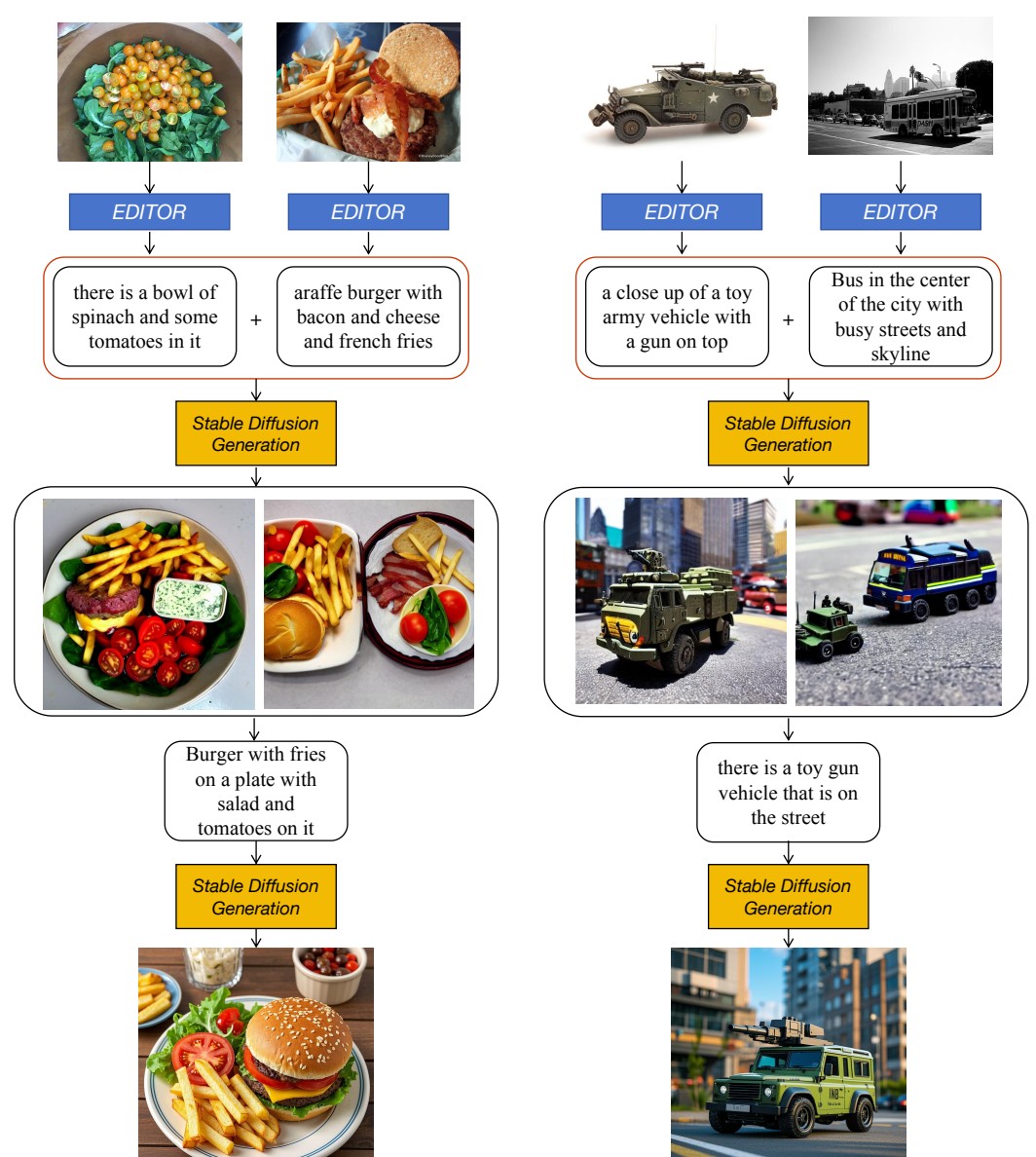

Figure 12: Application of evolutionary multi-concept generation.

Table 11: Impact of initial text length.

| Metric | 12 tokens | 16 tokens | 20 tokens |
|---|---|---|---|
| CLIP Score↑ | 0.785 | 0.789 | 0.783 |
| LPIPS Score↓ | 0.496 | 0.413 | 0.406 |
| Prompt Precision↑ | 0.852 | 0.863 | 0.884 |
| Prompt Recall↑ | 0.864 | 0.886 | 0.909 |
| Prompt F1↑ | 0.858 | 0.873 | 0.895 |
| Prompt PPL↓ | 79.940 | 79.234 | 80.338 |

Table 12: Impact of interation.

| Metric | 500 | 1000 | 1500 | 2000 |
|---|---|---|---|---|
| CLIP Score ↑ | 0.784 | 0.796 | 0.799 | 0.802 |
| LPIPS Score↓ | 0.456 | 0.414 | 0.460 | 0.413 |
| Prompt Precision↑ | 0.899 | 0.900 | 0.895 | 0.893 |
| Prompt Recall↑ | 0.919 | 0.920 | 0.916 | 0.916 |
| Prompt F1 ↑ | 0.908 | 0.908 | 0.904 | 0.903 |
| Prompt PPL ↓ | 70.364 | 80.659 | 83.457 | 87.907 |

improves the similarity between the generated image and the target image, which is the goal of our optimization. However, the improvement slows down over time, meaning that additional iterations result in smaller gains. Additionally, with more iterations, the quality of the inverted prompt decreases. This happens because over-optimization introduces complex or unclear tokens and semantic structures, making the prompt harder to interpret. To balance image similarity, prompt quality, and computational cost, we set the default number of iterations to 1000.

**Impact of the Number of Denoising Step.** EDITOR can invert the prompts for text-to-image diffusion models with small denoising steps. We explore the impact of denoising step number from 5

Table 13: Impact of the number of denoising step.

| Metric | 5 denoising steps | 10 denoising steps | 15 denoising steps |
|---|---|---|---|
| CLIP Score↑ | 0.796 | 0.792 | 0.792 |
| LPIPS Score↓ | 0.414 | 0.418 | 0.406 |
| Prompt Precision↑ | 0.900 | 0.897 | 0.894 |
| Prompt Recall↑ | 0.920 | 0.915 | 0.915 |
| Prompt F1↑ | 0.908 | 0.904 | 0.903 |
| Prompt PPL↓ | 80.659 | 98.156 | 65.881 |

Table 14: Impact of image captioning models.

| Metric | GIT-Large | BLIP-Large | BLIP2-OPT-2.7B |
|---|---|---|---|
| CLIP Score↑ | 0.763 | 0.796 | 0.781 |
| LPIPS Score ↓ | 0.427 | 0.414 | 0.467 |
| Prompt Precision↑ | 0.890 | 0.900 | 0.882 |
| Prompt Recall↑ | 0.892 | 0.920 | 0.906 |
| Prompt F1↑ | 0.889 | 0.908 | 0.894 |
| Prompt PPL↓ | 213.565 | 80.659 | 98.681 |

Table 15: Impact of random noise initialization.

| Metric | 0 | 34 | 258 | 423 |
|---|---|---|---|---|
| CLIP Score↑ | 0.784 | 0.770 | 0.785 | 0.779 |
| LPIPS Score↓ | 0.434 | 0.442 | 0.433 | 0.431 |
| Prompt Precision↑ | 0.884 | 0.907 | 0.891 | 0.897 |
| Prompt Recall↑ | 0.907 | 0.913 | 0.901 | 0.909 |
| Prompt F1↑ | 0.894 | 0.908 | 0.893 | 0.900 |
| Prompt PPL↓ | 45.955 | 59.683 | 92.843 | 75.296 |

Table 16: Impact of beam search size.

| Metric | 1 | 4 |
|---|---|---|
| CLIP Score↑ | 0.832 | 0.826 |
| LPIPS Score↓ | 0.439 | 0.414 |
| Prompt Precision↑ | 0.830 | 0.841 |
| Prompt Recall↑ | 0.823 | 0.824 |
| Prompt F1Score↑ | 0.826 | 0.832 |
| Prompt PPL↓ | 108.629 | 187.299 |

to 15 steps. As shown in Table 13, the difference in performance across different step numbers is relatively small, indicating that the number of denoising steps has limited influence on the image similarity and the quality of prompts. However, using fewer steps, such as 5 steps, significantly reduces optimization complexity and computational cost, improving overall efficiency. In Stable Diffusion, denoising steps below 5 often lead to insufficient noise reduction, which impacts image quality. Therefore, we set the default number of denoising step to 5.

**Impact of Different Image Captioning Models.** To justify the image captioning model choices for the prompt initialization, we show the results in Table 14 using three state-of-the-art open-source models: GIT-large (Wang et al., 2022), BLIP-large (Li et al., 2022a) and BLIP2-OPT-2.7B (Li et al., 2022b). Among the models, GIT-large shows the weakest performance. Notably, BLIP2-OPT-2.7b leverages a large language model, resulting in more sophisticated initialized prompts that achieve better textual alignment with the ground-truth. BLIP-large achieves better image quality with higher CLIP Similarity. We select BLIP-large as our default image captioning model. It is more lightweight compared to BLIP2-OPT-2.7b while still maintaining excellent performance.

**Impact of Random Noise Initialization.** The choice of random noise seed in diffusion models significantly affects the output. We further investigated the influence of seed values used for initializing random noise on the results of our method. As shown in Table 15, the impact is minimal. This demonstrates the robustness of our approach and the ability of the model to effectively manage and mitigate the effects of random noise during the inversion process.

**Impact of Beam Search Size in E2T model.** We further analyze the effect of the beam search size in the embedding-to-text decoding stage. As shown in Table 16, different beam sizes lead to only marginal variations across similarity, alignment, and perplexity metrics. This demonstrates that the beam search setting has a negligible effect on our final results, confirming the stability of our method with respect to decoding choices. We set the default beam search size in the embedding-to-text decoding stage to 4.

# F STATISTICAL ANALYSIS

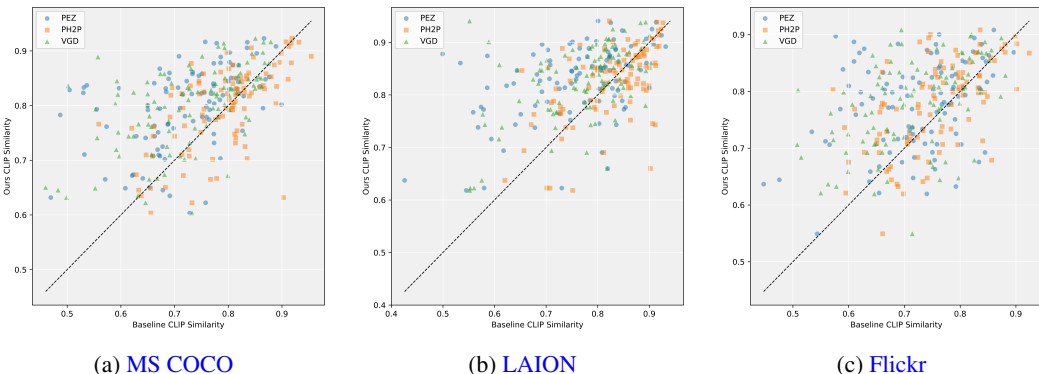

|                    |                   |                    |
|:------------------:|:-----------------:|:------------------:|
| (a) MS COCO        | (b) LAION         | (c) Flickr         |

Figure 13: Per-image CLIP similarity scatter plots comparing EDITOR with three baselines on different datasets.

We provide a detailed statistical analysis based on our core metric, CLIP similarity, on the Flickr subset. As shown in Table 17, EDITOR achieves both the highest mean CLIP similarity and the tightest confidence intervals among all methods. The paired $t$-tests in Table 18 further indicate that the improvements over all baselines are statistically significant (all $p < 0.01$).

Table 17: Mean CLIP similarity, 95% confidence intervals (CI), and standard error of the mean (SEM) on Flickr.

| Method | Mean CLIP ↑ | 95% CI | SEM |
|---|---|---|---|
| **EDITOR (Ours)** | **0.7762** | **[0.7607, 0.7918]** | **0.0078** |
| PEZ | 0.7217 | [0.7040, 0.7395] | 0.0090 |
| PH2P | 0.7548 | [0.7392, 0.7703] | 0.0078 |
| VGD | 0.7146 | [0.6965, 0.7326] | 0.0091 |

Table 18: Paired $t$-test statistics comparing EDITOR against baselines on Flickr (CLIP similarity).

| Compared Baseline | $t$-statistic | $p$-value |
|---|---|---|
| PEZ | 5.5480 | $2.41 \times 10^{-7}$ |
| PH2P | 2.9483 | $3.99 \times 10^{-3}$ |
| VGD | 7.1118 | $1.81 \times 10^{-10}$ |

We further visualize per-image CLIP similarity using scatter plots. As shown in Figure 13, across all three baselines (PEZ, PH2P, VGD), the majority of points lie above the diagonal, indicating that EDITOR achieves higher CLIP similarity on a per-image basis. The spread of points further shows that our method consistently improves both the mean and the worst-case performance, with fewer severe outliers compared to existing approaches, confirming that the gains are not driven by a small subset of images.

# G THE USE OF LARGE LANGUAGE MODELS (LLMS)

In preparing this paper, we employed a large language model (LLM) to aid in editing and polishing the writing. The LLM was used to improve clarity, grammar, and academic style, as well as to suggest alternative phrasings for certain sections such as the contributions, ablation study descriptions, and reproducibility statement. The technical content, research ideas, experiments, and analyses were fully developed and conducted by the authors. No LLM was used to generate experimental results, implement algorithms, or design the methodology.

# H ADDITIONAL RESULTS

Table 19: Additional examples of prompt inversion

| Original Prompt | Inverted Prompt |
| --- | --- |
| a photograph of an astronaut riding a horse | an astronaut rides on a horse in space |
| An ancient ruins overgrown with lush greenery | Ancient stone ruins standing in lush green, arid setting |
| A group of people taking photographs with cellphones | A bunch of people are taking selfies with their cell phones |
| two cows sitting on some dirt and grass | You can see that there are two cows standing in the grass |
| A slice of pie on top of a white plate near a fork | there's a piece of cake on a plate with a fork |
| Crochet Baby Bunny Ear Hat and Bootie Set - Light Blue | crochet bunny hat and booties |

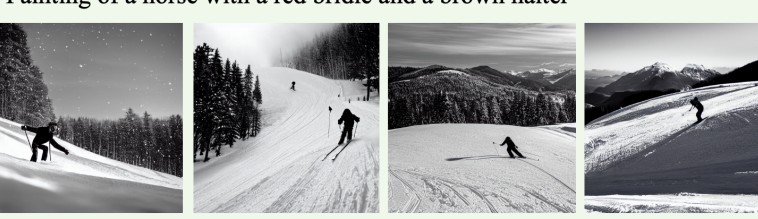

Poster for the movie Zootopia

araffe burger with bacon and cheese and french fries

Painting of a horse with a red bridle and a brown halter

skier going down a snowy hill in black and white picture

Figure 14: Additional images with invert prompts

