# OpenReview forum: "EDITOR: Effective and Interpretable Prompt Inversion for Text-to-Image Diffusion Models"
_ICLR.cc/2026/Conference — Submitted to ICLR 2026_

### Official Review · Reviewer_zXzv · 2025-10-28

**Soundness:** 2
**Presentation:** 1
**Contribution:** 2
**Rating:** 4
**Confidence:** 4

**Summary:**

This paper proposes EDITOER for inverting prompts in text-to-image diffusion models. Specifically, EDITOR first initializes a prompt using image captioning model, then optimize the text embedding of the initialized prompt based on the image reconstruction, and finally applies a pretrained embedding-to-text model to retrieve the prompt. Experiments show that the proposed method improves image similarity, textual alignment, and prompt interpretability, and can be effectively applied to various applications.

**Strengths:**

1.	The paper clearly points out the challenge of current prompt inversion and proposes an effective solution.
2.	Extensive experiments validate the advancement of the proposed method. The application study is interesting and further shows the potential of the method.

**Weaknesses:**

1.	Reverse-engineering for latent text embedding has been explored in previous study [1], but is not discussed in this work. Also, it would also be interesting to see the effect of initialization with the DDIM null text inversion mentioned in [1], which is widely used in the text-to-image model for maintaining the image content.
2.	The key technical innovation locates in the embedding-to-text model, but it is based on a preliminary work, not newly proposed here. So, the technical innovation of this paper is marginal.
3.	The description of the embedding inversion is unclear. (1) It needs to train two E2T models, $M_{corr}$ and $M_{zero}$?  (2) The training details of embedding-to-text model are missing, e.g., the number of training text-representation pairs. (3) The details of beam search.
4.	The paper lacks an ablation study comparing the use of $M_{zero}$ only with using $M_{zero}$ and $M_{corr}$.
5.	Line 200 mentions “attacker”, but this term is neither explained nor referenced elsewhere in the paper.
6.	There are errors/typos in the paper. (1) Line 284 $T(y)$ (2) Line 836, wrong description for Figure 9.

**Questions:**

See the weakness.

---

> ### Author Response · Authors · 2025-11-22
> **Reply to Reviewer zXzv (1/2)**
>
> We appreciate the valuable feedback from the reviewer, which has been very helpful in improving our paper. Based on your feedback, we have revised the paper as follows. Please let us know if this resolves your concerns.
>
> >***[W1]: Reverse-engineering for latent text embedding has been explored in previous study [1], but is not discussed in this work. Also, it would also be interesting to see the effect of initialization with the DDIM null text inversion mentioned in [1], which is widely used in the text-to-image model for maintaining the image content.***
>
> Could the reviewer kindly clarify which specific work is referred to as “[1]”? The citation was not provided in the review. We would greatly appreciate confirmation so that we can address the correct reference in the revision.
>
>
>
> > ***[W2]: The key technical innovation locates in the embedding-to-text model, but it is based on a preliminary work, not newly proposed here. So, the technical innovation of this paper is marginal.***
>
> We thank the reviewer for the comment. We agree that the E2T module is not itself a novel architecture. However, **E2T is only one component of our framework and not the primary contribution.** The novelty lies in the **overall pipeline design and how E2T is used**, not in the model architecture. In EDITOR:
> - The core innovation is the **three-stage problem decomposition** (Initialize → Optimize → Decode).
> - The E2T model plays the role of a **semantic interface**, enabling the contextual embedding to be optimized independently of textual decoding.
> - It is the design, not the embedding-to-text model itself that allows EDITOR to reconstruct optimized embeddings with high fidelity and then convert them to natural language while maintaining semantic alignment.
>
> This is fundamentally different from PEZ/PH2P, where the final text is tightly coupled to discrete optimization and vocabulary projection. Our framework **decouples embedding optimization from decoding**, enabling behaviors that previous methods fundamentally cannot achieve (e.g., low PPL, stable embedding inversion, plug-and-play caption-based initialization).
> Thus, while the E2T model is built upon an existing language model, **its role in our new problem formulation and pipeline is the true contribution**, rather than the model architecture itself.
>
>
>
> > ***[w3]: The description of the embedding inversion is unclear. (1) It needs to train two E2T models? (2) The training details of embedding-to-text model are missing, e.g., the number of training text-representation pairs. (3) The details of beam search.***
>
> Yes, we do train 2 models: $M_{zero}$ and $M_{corr}$, which together constitute our E2T. $M_{zero}$ takes the optimized contextual embedding as input and produces a coarse initial text hypothesis. The correction model $M_{corr}$ then refines this hypothesis to pull the decoded prompt closer to the optimized embedding.
>
> Second, regarding training details of the embedding-to-text model, we train on the MSMARCO corpus, which contains **8.84M text–representation pairs**. Other training details were already included in Appendix C of our original submission. Specifically, the backbone is T5-base, with batch size 32, learning rate 1e-3, maximum token length 32, and training for 60 epochs for $M_{zero}$  and 35 epochs for $M_{corr}$.
>
> Finally, our E2T model is instantiated as a T5-base language model, so at decoding time we follow standard seq2seq practice and use beam search to explore multiple candidate prompt sequences. Our default beam size is 4. We additionally conducted an ablation study with beam size set to 1 (i.e., greedy decoding) and found the results to be very close to those with a beam size of 4, indicating that the beam setting has only a negligible effect on our final results. The results are as follows:
>
>
> | Beam Size | CLIP Score ↑ | LPIPS Score ↓ | Precision ↑ | Recall ↑ | F1 ↑  | PPL ↓   |
> |-----------|-------------------|--------------------|---------------------|------------------|--------------------|---------------|
> | 1(greedy decoding)         | 0.832             | 0.439              | 0.830               | 0.823            | 0.826              | 108.629       |
> | 4         | 0.826             | 0.414              | 0.841               | 0.824            | 0.832              | 187.299       |
>
>
> We have clarified these points and added the above details in the revised manuscript.

---

> > ### Author Response · Authors · 2025-11-22
> > **Reply to Reviewer zXzv (2/2)**
> >
> > > ***[w4]: The paper lacks an ablation study of the zero model and the correction model.***
> >
> > we conduct an ablation study that isolates the contribution of the zero-step model and the correction model. As shown in the table below, using only the zero-step model already yields strong performance, but adding the correction Model consistently improves key metrics across all four datasets.
> >
> > Dataset|  | CLIP Score ↑ | LPIPS Score ↓ | Precision ↑ | Recall ↑ | F1 ↑  | PPL ↓   |
> > | -------- | -------- | -------- | -------- | -------- | -------- | -------- |-------- |
> > |MS COCO | Zero-Step Model  |0.772 |0.414 |0.898 |0.910 |0.903 |105.906 |
> > | | Zero-Step Model + Correction Model |0.796| 0.414| 0.900| 0.920| 0.908| 80.659  |
> > |LAION | Zero-Step Model  | 0.811| 0.391|0.832 |0.820 |0.826 | 214.636|
> > | | Zero-Step Model + Correction Model |0.826 |0.401| 0.841| 0.824| 0.832| 187.299  |
> > |Flickr | Zero-Step Model  |0.764 |0.402 |0.885 |0.913 |0.896 |100.287 |
> > | | Zero-Step Model + Correction Model |0.776  |0.424 |0.892 | 0.915 |0.900| 90.320  |
> > |DiffusionDB | Zero-Step Model  |0.803 |0.412 | 0.829|0.817 |0.822 |110.294 |
> > | | Zero Model + Correction Model |0.807 |0.385 |0.835 |0.813 |0.823 |105.185 |
> >
> > > ***[w5-w6]: Line 200 mentions “attacker”, but this term is neither explained nor referenced elsewhere in the paper. There are errors/typos in the paper. (1) Line 284 (2) Line 836, wrong description for Figure 9.***
> >
> > We appreciate the reviewers' corrections. We have changed "attacker" to "engineer" in the revised version, revised the notation errors in lines 284 and revised the description of Figure 9 in the updated version.

---

> > > ### Comment · Reviewer_zXzv · 2025-11-24
> > > **Thanks for the responses.**
> > >
> > > Apologies for the earlier omission. The referenced work [1] is provided below.
> > >
> > > After reading the authors’ response and the other reviewers’ comments, I still believe the main issue of this paper lies in its novelty, as also noted by two reviewers. I understand that the authors aim to claim novelty in their proposed initialize–optimize–decode pipeline, and that the key advantage is avoiding vocabulary projection and discrete optimization via using the "continuous contextual embedding space after the text encoder.". However, employing a continuous embedding space after the text encoder is already a commonly used practice in related domains [1, 2].
> > >
> > > [1] Dong, W., Xue, S., Duan, X., & Han, S. (2023). Prompt tuning inversion for text-driven image editing using diffusion models. In Proceedings of the IEEE/CVF International Conference on Computer Vision (pp. 7430-7440).
> > > [2] Dalva, Y., & Yanardag, P. (2024). Noiseclr: A contrastive learning approach for unsupervised discovery of interpretable directions in diffusion models. In Proceedings of the IEEE/CVF conference on computer vision and pattern recognition (pp. 24209-24218).

---

> > > > ### Author Response · Authors · 2025-11-25
> > > > **Clarification Response Regarding Novelty**
> > > >
> > > > Dear Reviewer zXzv,
> > > >
> > > > We thank the reviewer for the follow-up comment and providing the missing citation. Although continuous embedding optimization has been used in image-editing works such as Prompt Tuning Inversion [1], NoiseCLR [2], and Null-Text Inversion [3], these methods operate with assumptions fundamentally different from those required for prompt inversion. Editing approaches optimize free-form, timestep-dependent embeddings that preserve DDIM trajectories or adjust conditioning signals. **Such embeddings encode per-step diffusion dynamics rather than linguistic semantics, are unconstrained by the manifold of valid textual prompts, and cannot be shared across timesteps or decoded into human-readable text.** For example, [1] optimizes a sequence of timestep-specific conditional embeddings, [2] searches for directions in noise space rather than text-embedding space, and [3] adjusts null-text embeddings independently per diffusion step. None of these representations correspond to a single prompt, making direct adaptation to prompt inversion impossible.
> > > >
> > > >
> > > > In contrast, prompt inversion imposes substantially stronger requirements: the goal is to recover a single global contextual embedding that conditions the entire diffusion process, reconstructs the image at the pixel level, and remains sufficiently close to the prompt manifold to be decoded into natural language. **This setting reflects the standard user-facing usage scenario of text-to-image models and defines the task we aim to solve.** Achieving this required us to optimize the global contextual embedding and to use an embedding-to-text decoder capable of mapping contextualized CLIP embeddings back to fluent text. These capabilities entirely absent in prior editing-based methods [1–3], since their embeddings are never intended to be interpretable. Our caption-based initialization also provides a linguistically meaningful and semantically grounded starting point, ensuring that the optimization remains aligned with interpretable prompt space rather than drifting into free-form latent directions.
> > > >
> > > > Beyond these structural differences, the optimization objective of prompt inversion diverges sharply from editing. Methods such as [1] and [3] aim to match or stabilize DDIM trajectories to preserve editing consistency, which is a local, stepwise constraint. Our objective, however, is to directly minimize the full image reconstruction error of the diffusion model with respect to the contextual embedding. **This end-to-end formulation operates at the global image level rather than local trajectory level and is inherently sampler-agnostic, functioning across DDPM, DDIM, and DiT-based generation pipelines.** As shown in **Table 6**, this global optimization achieves consistent reconstruction performance across SD1.5, SDXL, and SD3.5, highlighting that our approach addresses a different and significantly more challenging problem than editing-based embedding methods.
> > > >
> > > > **For these reasons, EDITOR is not a straightforward application of existing techniques.** It is the **first method** to make continuous embedding optimization compatible with the strict linguistic, structural, and global reconstruction requirements of prompt inversion, and **the first to close the loop from image → contextual embedding → human-readable text.** This defines a new problem formulation and contributes a novel technical pipeline beyond the scope of prior continuous-embedding editing methods [1–3].
> > > >
> > > > [1] Dong, W., Xue, S., Duan, X., & Han, S. (2023). Prompt tuning inversion for text-driven image editing using diffusion models. In Proceedings of the IEEE/CVF International Conference on Computer Vision (pp. 7430-7440).
> > > >
> > > > [2] Dalva, Y., & Yanardag, P. (2024). Noiseclr: A contrastive learning approach for unsupervised discovery of interpretable directions in diffusion models. In Proceedings of the IEEE/CVF conference on computer vision and pattern recognition (pp. 24209-24218).
> > > >
> > > > [3] Mokady, R., Hertz, A., Aberman, K., Pritch, Y., & Cohen-Or, D. (2023). Null-text inversion for editing real images using guided diffusion models. In Proceedings of the IEEE/CVF conference on computer vision and pattern recognition (pp. 6038-6047).

---

### Official Review · Reviewer_SgN7 · 2025-10-30

**Soundness:** 2
**Presentation:** 3
**Contribution:** 2
**Rating:** 6
**Confidence:** 3

**Summary:**

The paper introduces EDITOR, a prompt‑inversion pipeline for text‑to‑image diffusion models that aims to recover readable prompts that reliably reproduce a target image. EDITOR initializes from an image‑captioning prompt, optimizes the text encoder’s contextual embedding in continuous space to match the target image under a fixed seed, and converts the optimized embedding back to text with an embedding‑to‑text (E2T) model plus a small correction module.

**Strengths:**

1. Optimizing contextual embeddings after the transformer and deferring discrete text decoding to an E2T model is novel

2. EDITOR improves image similarity and prompt interpretability/text alignment.

**Weaknesses:**

1. EDITOR depends on a trained E2T module;  this adds implementation and computation costs.

2. Mapping embedding to text to embedding may not be strict,  paraphrases can drift semantics. The extent to which this harms re-generation fidelity and editability is under-measured. Authors are suggested to give more details.

3. Experiments focus on COCO/LAION/Flickr subsets.The scale of the dataset is relatively limited.

**Questions:**

1. The pipeline introduces a trained embedding-to-text (E2T) model and a correction module, increasing complexity. The paper gives limited profiling of training time, memory, and sample efficiency.

2. What about the performance on more datasets?

---

> ### Author Response · Authors · 2025-11-22
> **Reply to Reviewer SgN7 （1/2)**
>
> We appreciate the valuable feedback from the reviewer, which has been very helpful in improving our paper. Based on your feedback, we have revised the paper as follows. Please let us know if this resolves your concerns.
>
> > ***[W1, Q1]: EDITOR depends on a trained E2T module; this adds implementation and computation costs. The pipeline introduces a trained embedding-to-text (E2T) model and a correction module, increasing complexity. The paper gives limited profiling of training time, memory, and sample efficiency.***
>
> We appreciate the reviewer’s comment regarding the additional implementation and computation cost introduced by the E2T module. In our pipeline, the E2T model requires a one-time, offline training stage, which takes 2 days on 8×A100 GPUs. This cost is incurred only once, and the community does not need to retrain the model, as we provide the fully pre-trained checkpoint. At inference time, the overhead is lightweight. Loading the pre-trained E2T model adds only **2.45 GB** of memory. During inference, it uses about **7.31 GB**, all in float32 precision. The actual embedding-to-text decoding step, i.e., converting a reconstructed embedding into a prompt, takes just **7.36 seconds** on a single NVIDIA A100 GPU. Compared to the preceding embedding optimization phase and iterative baselines such as PH2P, which takes 3482s, this overhead is negligible.
>
> Thus, while E2T introduces an additional component, its training cost is a one-time offline investment, and its **inference-time overhead in both memory and runtime is minimal**.
>
>
> > ***[W2]: Mapping embedding to text to embedding may not be strict, paraphrases can drift semantics. The extent to which this harms re-generation fidelity and editability is under-measured. Authors are suggested to give more details.***
>
> We thank the reviewer for raising this important point. We agree that naively mapping embedding to text to embedding can introduce semantic drift, especially when using simple nearest-neighbor vocabulary projection as in PEZ/PH2P. However, **EDITOR is explicitly designed to avoid this problem**, and our paper presents several mechanisms and empirical results demonstrating that semantic drift is minimal and does not harm reconstruction fidelity or editability.
>
> **1. Continuous contextual-embedding optimization avoids projection drift**
>     As shown in Figure 2 and Figure 4, EDITOR never operates in the token-embedding space where projection is required. Instead, we optimize contextual embeddings after the transformer, in a continuous latent space. This eliminates the main source of semantic drift present in prior work: vocabulary projection. Table 2 shows that vocabulary projection in PEZ/PH2P causes extreme embedding discrepancy (cosine similarity only 0.167). Our embedding inversion achieves 0.737 cosine similarity, a 4.4× improvement. Thus, the optimized embedding remains close to the true optimum and is not pushed far away during decoding.
>
> **2. The E2T model is trained inside the diffusion model’s embedding distribution**
>     Our embedding-to-text (E2T) model is trained directly on the diffusion model’s own text-encoder embeddings. We construct a large corpus of (prompt, embedding) pairs from the same encoder and train E2T to recover prompts whose embeddings closely match the targets. As shown in Table 9, this E2T module achieves 0.968 precision, 0.971 recall, and 0.969 F1, indicating highly faithful embedding reconstruction. In practice, this joint design keeps the decoded prompts tightly aligned with the optimized embeddings, so paraphrasing drift is minimal and does not noticeably harm re-generation fidelity or editability.
>
> **3. Empirical evidence: no drift in fidelity or editability**
> We measure the effect of embedding inversion directly on downstream tasks:
> - **High reconstruction fidelity (CLIP/LPIPS)**
> Across all datasets (Table 3, page 6), EDITOR achieves the highest CLIP similarity and lowest LPIPS among all baselines, showing that embedding → text → embedding does not degrade image reconstruction.
> - **High textual alignment (BERTScore)**
> EDITOR’s decoded prompts have the highest precision/recall/F1, indicating that the semantic content is not lost during inversion.
> - **Editability and concept grounding remain intact**
> Applications such as object removal/replacement (Figure 6, page 9) and unsupervised segmentation (Figure 11, page 17) rely on token-level grounding. The success of these tasks demonstrates that paraphrasing drift does not degrade semantic editability.

---

> ### Author Response · Authors · 2025-11-22
> **Reply to Reviewer SgN7 （2/2)**
>
> > ***[W3, Q2]: Experiments focus on COCO/LAION/Flickr subsets.The scale of the dataset is relatively limited. What about the performance on more datasets?***
>
> We additionally evaluate EDITOR on a new and much broader dataset, **DiffusionDB [1]**, and compare it against all baselines under the same protocol. Different from MS COCO, LAION, and Flickr, DiffusionDB consists of real prompts and images generated by diverse users, spanning a wide range of visual styles, prompt structures, and noise levels. This leads to an **image distribution that is clearly different from the curated datasets used earlier**. Despite this significant distribution shift, **EDITOR continues to outperform PEZ, PH2P, and VGD across all metrics on DiffusionDB**, as shown below:
> | Method | CLIP Score ↑ | LPIPS Score ↓ | Precision ↑ | Recall ↑ | F1 ↑  | PPL ↓   |
> | -------- | -------- | -------- | -------- | -------- | -------- |-------- |
> | PEZ |0.766 |0.466 |0.800 |0.807 |0.803 |10,896.730 |
> | PH2P |0.742 |0.459 |0.766 |0.795 |0.780 |7,286.705 |
> | VGD |0.780 |0.454 |0.806 |0.814 |0.810 |553.287 |
> | EDITOR |0.807 |0.385 |0.835 |0.813 |0.823 |105.185 |
>
> [1] Wang et al.,  "DiffusionDB: A Large-scale Prompt Gallery Dataset for Text-to-Image Generative Models," ACL 2023.

---

### Official Review · Reviewer_H81s · 2025-10-31

**Soundness:** 3
**Presentation:** 3
**Contribution:** 2
**Rating:** 6
**Confidence:** 4

**Summary:**

The paper introduces EDITOR, a prompt-inversion pipeline for text-to-image diffusion models that aims to produce prompts that are interpretable and effective at re-generating the target image. It (i) initializes from a caption, (ii) optimizes the contextual text embedding directly in the encoder’s continuous space to reconstruct the given image, and (iii) maps that optimized embedding back to fluent text via an embedding-to-text (E2T) model with a small beam-search correction. Experiments on COCO/LAION/Flickr and transfer tests (e.g., SDXL-Turbo, SD3.5-Medium) report consistent gains vs. PEZ/PH2P and caption-only baselines.

**Strengths:**

- The paper provides a clean, modular pipeline that others can readily reuse.
- Better similarity and more fluent prompts than PEZ/PH2P and captioners across multiple datasets and model variants.
- Produces prompts that are human-interpretable, aiding provenance/attribution and even downstream editing.

**Weaknesses:**

- The method composes established components; the main idea (optimize contextual embeddings, then decode to text) is a practical tweak rather than a new paradigm.
- Only 100 images per dataset is used for evaluation; it's unclear how stable gains are across broader distributions.
- Protocol choices (initialization, token/step budgets) could affect PEZ/PH2P competitiveness; a standardized compute budget table would be good to have.

**Questions:**

1. Add confidence intervals/paired tests and per-image scatter plots for key tables to show variance/outliers.
2. Provide per-stage cost (init, inversion by iteration, etc.), plus scaling with prompt length and denoising steps.
3. Provide sensitivity analysis to noise seeds and to the choice/mixture of caption initializers.

---

> ### Author Response · Authors · 2025-11-22
> **Reply to Reviewer H81s (1/3)**
>
> We appreciate the valuable feedback from the reviewer, which has been very helpful in improving our paper. Based on your feedback, we have revised the paper as follows. Please let us know if this resolves your concerns.
>
> >***[W1]: The method composes established components; the main idea (optimize contextual embeddings, then decode to text) is a practical tweak rather than a new paradigm.***
>
> While EDITOR is indeed modular, its contribution is not in introducing new components but in demonstrating that **a different pipeline enables capabilities that prior paradigms fundamentally lack**. Existing baselines (PEZ, PH2P) operate directly in token space and cannot benefit from stronger initialization: even when initialed with captioner outputs, they quickly converge to uninterpretable high-perplexity token sequences (**Table 7**).
> In contrast, our Initialize → Optimize → Decode framework forms a pipeline where each stage is fully differentiated and contributes uniquely:
>
> - *Initialization* meaningfully improves optimization stability and final quality—an effect absent in baselines.
> - *Embedding optimization* occurs in contextual latent space, avoiding the discrete bottlenecks that make PEZ/PH2P unstable.
> - *Decoding via E2T* produces fluent, human-readable prompts while preserving embedding fidelity.
>
> The fact that prior paradigms cannot exploit initialization or maintain interpretability highlights that EDITOR is not simply a tweak, but a functionally distinct paradigm. The resulting gains in CLIP/LPIPS/F1 across all datasets and architectures reflect benefits that cannot be reproduced by assembling the same components within the prior framework.
>
>
>
> > ***[W2]: Only 100 images per dataset is used for evaluation; it's unclear how stable gains are across broader distributions.***
>
> We additionally evaluate EDITOR on a new and much broader dataset, **DiffusionDB [1]**, and compare it against all baselines under the same protocol. Different from MS COCO, LAION, and Flickr, DiffusionDB consists of real prompts and images generated by diverse users, spanning a wide range of visual styles, prompt structures, and noise levels. This leads to an **image distribution that is clearly different from the curated datasets used earlier**. Despite this significant distribution shift, **EDITOR continues to outperform PEZ, PH2P, and VGD across all metrics on DiffusionDB**, as shown below:
> | Method | CLIP Score ↑ | LPIPS Score ↓ | Precision ↑ | Recall ↑ | F1 ↑  | PPL ↓   |
> | -------- | -------- | -------- | -------- | -------- | -------- |-------- |
> | PEZ |0.766 |0.466 |0.800 |0.807 |0.803 |10,896.730 |
> | PH2P |0.742 |0.459 |0.766 |0.795 |0.780 |7,286.705 |
> | VGD |0.780 |0.454 |0.806 |0.814 |0.810 |553.287 |
> | EDITOR |0.807 |0.385 |0.835 |0.813 |0.823 |105.185 |
>
> [1] Wang et al.,  "DiffusionDB: A Large-scale Prompt Gallery Dataset for Text-to-Image Generative Models," ACL 2023.
>
> > ***[W3]: Protocol choices (initialization, token/step budgets) could affect PEZ/PH2P competitiveness; a standardized compute budget table would be good to have.***
>
>
>
> We thank the reviewer for the suggestion. To ensure a fair comparison, all methods were evaluated under the same protocol settings. Specifically,
> - we set the prompt token budget to **16** for every approach.
> - Regarding initialization, Table 8 shows that even when PEZ and PH2P are initialized with prompts from an image captioning model, their performance **changes only marginally**. This is due to their optimization-with-vocabulary-projection mechanism: even if they start from interpretable prompts, the optimization trajectory quickly replaces them with high-perplexity token combinations. For this reason, we did not apply initialization to PEZ or PH2P in the main baseline experiments.
> - For the optimization budget, we allocated **1000** query budgets to all optimization-based approach, ensuring that PEZ, PH2P, and our method operate under the same step budget.

---

> ### Author Response · Authors · 2025-11-22
> **Reply to Reviewer H81s (2/3)**
>
> > ***[Q1]: Add confidence intervals/paired tests and per-image scatter plots for key tables to show variance/outliers.***
>
> We conduct the requested statistical analysis based on our core metric, **CLIP similarity** on **Flickr**. As shown in the tables below, EDITOR achieves both the **highest mean performance** and **tight confidence intervals** across all methods, and the paired t-tests indicate statistically significant improvements over every baseline (all p < 0.001).
>
>
> | Method | Mean CLIP Similarity | 95% Confidence Intervals  | Standard Error of the Mean    |
> |--------|-------------:|-------------------|--------|
> | Ours   | **0.7762**       | **[0.7607, 0.7918]**  | **0.0078** |
> | PEZ    | 0.7217       | [0.7040, 0.7395]  | 0.0090 |
> | PH2P   | 0.7548       | [0.7392, 0.7703]  | 0.0078 |
> | VGD    | 0.7146       | [0.6965, 0.7326]  | 0.0091 |
>
> | Compared Baseline | t-statistic | p-value      |
> |----------|------------:|-------------:|
> | PEZ      |  5.5480     | 2.4137e-07   |
> | PH2P     |  2.9483     | 3.9857e-03   |
> | VGD      |  7.1118     | 1.8130e-10   |
>
> In the revised version, we also include per-image scatter plots in **Figure 13** to visualize point-wise differences, variance, and potential outliers. Across all three baselines (PEZ, PH2P, VGD), the majority of points lie above the diagonal, indicating that **EDITOR achieves higher CLIP similarity on a per-image basis**. The spread of points also shows that our method consistently improves both the mean and the worst-case performance, with fewer severe outliers compared to existing approaches. These plots visually confirm that the gains reported in the tables are stable across individual images rather than driven by a small subset.
>
> > ***[Q2]: Provide per-stage cost (init, inversion by iteration, etc.), plus scaling with prompt length and denoising steps.***
>
> We run all experiments on an **A100-SXM4-80GB GPU** and use **float32 precision**.
>
> For the initialization stage, we use BLIP-Large as the captioning model. Generating the initial prompt takes only **1.48 seconds**. During this stage, loading the BLIP-Large model and running inference require approximately **8 GB** of GPU memory.
>
> For the inversion-by-iteration phase, running 1000 optimization steps takes **1357.30 seconds** (i.e., 1.35 s per iteration).
> This optimization stage involves storing gradients, activations, and optimizer states, which significantly increases memory consumption. As a result, this phase requires approximately **32 GB** of GPU memory.
>
> For the embedding inversion phase, the step of using the pre-trained E2T model to convert the final embedding into text is lightweight and takes only **7.36 seconds**.
> Loading and running inference for the E2T model require approximately **7 GB** of GPU memory.
>
> In our implementation, the prompt length has a negligible effect on computational cost across all stages. In contrast, increasing the number of denoising steps has a clear impact on both runtime and memory consumption. For example, with 5 denoising steps, a single inversion iteration takes **1.35 seconds**, whereas increasing to 10 denoising steps raises the per-iteration time to **2.54 seconds**. This increase in denoising depth also leads to higher memory usage: the per-iteration GPU memory footprint grows from approximately **32 GB** at 5 steps to around **36 GB** at 10 steps, primarily due to the additional activations and intermediate states that must be retained during the backward pass.

---

> ### Author Response · Authors · 2025-11-22
> **Reply to Reviewer H81s (3/3)**
>
> > ***[Q3]: Provide sensitivity analysis to noise seeds and to the choice/mixture of caption initializers***
>
> We appreciate the reviewer’s comment. We included sensitivity analyses on both (1) random noise seeds and (2) the choice of caption initializers in our ablation study (Tables 14 and 13 in the first submission). For clarity, we restate the results below.
>
> **Impact of random noise initialization.**
>
> | Seed  | CLIP Score ↑ | LPIPS Score ↓ | Precision ↑ | Recall ↑ | F1 ↑  | PPL ↓   |
> |-------|--------|---------|--------------------|------------------|-------------|--------------|
> | 0     | 0.784  | 0.434   | 0.884              | 0.907            | 0.894       | 45.955       |
> | 34    | 0.770  | 0.442   | 0.907              | 0.913            | 0.908       | 59.683       |
> | 258   | 0.785  | 0.433   | 0.891              | 0.901            | 0.893       | 92.843       |
> | 423   | 0.779  | 0.431   | 0.897              | 0.909            | 0.900       | 75.296       |
>
>
> We evaluates different random seeds for diffusion noise. All metrics vary only slightly across seeds, showing that our method is robust to noise initialization and that the optimization effectively mitigates stochasticity in the diffusion process.
>
>
> **Impact of different image captioning models.**
> | Seed  | CLIP Score ↑ | LPIPS Score ↓ | Precision ↑ | Recall ↑ | F1 ↑  | PPL ↓   |
> |----------------|--------|---------|--------------------|------------------|-------------|--------------|
> | GIT-Large      | 0.763  | 0.427   | 0.890              | 0.892            | 0.889       | 213.565      |
> | BLIP-Large     | 0.796  | 0.414   | 0.900              | 0.920            | 0.908       | 80.659       |
> | BLIP2-OPT-2.7B | 0.781  | 0.467   | 0.882              | 0.906            | 0.894       | 98.681       |
>
>
> We compare GIT-large, BLIP-large, and BLIP2-OPT-2.7B as caption initializers. GIT-large performs worst. BLIP2-OPT-2.7B gives better textual alignment (lower PPL), while BLIP-large achieves the best image quality (highest CLIP) with competitive text metrics. We therefore adopt BLIP-large as the default. As also illustrated in **Figure 8**, a stronger initializer leads to better EDITOR performance; BLIP-large yields the highest initialization scores, and thus the EDITOR results based on it are correspondingly better.

---

### Official Review · Reviewer_Kp1H · 2025-11-02

**Soundness:** 3
**Presentation:** 3
**Contribution:** 2
**Rating:** 4
**Confidence:** 5

**Summary:**

This paper introduces a method for generating fluent, human-readable prompts. The approach begins by initializing with prompts produced by an image captioning model, which serve as a starting point for optimizing embeddings within the continuous latent space of diffusion models. These optimized embeddings are then transformed back into natural language using an embedding-to-text model. The effectiveness of the proposed technique is demonstrated through experimental comparisons with PEZ and PH2P across the MS COCO, LAION, and Flickr datasets.

**Strengths:**

1. The work addresses an interesting problem of reverse engineering diffusion models.

2.Comprehensive evaluations and ablations show the effectiveness of the approach in comparison to prior work.

**Weaknesses:**

1.The work has limited novelty in the sense that it combines the gradient based optimization of prior work with the latent space of an existing model.

2.The notations and equations are incorrect. The cross-entropy loss and the MLE loss are not correctly defined in equation 4 and 6.

3. The approach does not consider recent multimodal architectures such as SD3.

4. Comparison to recent prompt inversion/search techniques such as [1].
[1] STEPS: Sequential Probability Tensor Estimation for Text-to-Image Hard Prompt  Search. CVPR 2025.

**Questions:**

1. Can the authors revisit and clarify the notations and equations. What is the difference between text decoder D and M.textdecoder in the algorithm. Also image is denoted by \mathbf{x} or $x$ at places.

2. How does the approach extend to embedding spaces of the more recent architectures like SD3? Can this be mapped to the embedding of the captioning model?

3. How does it compare to the recent work on prompt search which also generates human readable prompts?

---

> ### Author Response · Authors · 2025-11-22
> **Reply to Reviewer Kp1H (1/2)**
>
> We appreciate the valuable feedback from the reviewer, which has been very helpful in improving our paper. Based on your feedback, we have revised the paper as follows. Please let us know if this resolves your concerns.
>
> > ***[W1]: the work has limited novelty in the sense that it combines the gradient based optimization of prior work with the latent space of an existing model.***
>
> We thank the reviewer for the comment. Although EDITOR uses gradient-based optimization, its optimization **fundamentally differs** from prior work such as PEZ and PH2P, both conceptually and algorithmically.
>
> Prior approaches tightly **couple** gradient optimization with **vocabulary projection**, meaning that every gradient step must be projected back into the discrete token space. This coupling makes the optimization landscape highly unstable: each update is forced to snap to the nearest token embedding, causing oscillation, vanishing gradients, and ultimately **extremely low optimization efficiency**. As shown in Table 2 of our submission, this leads to very large embedding discrepancies (cosine similarity only 0.167) and high-perplexity prompts.
>
> In contrast, EDITOR introduces a different optimization formulation. We **completely decouple** gradient optimization from text decoding:
> - The optimization is performed **solely** in the continuous contextual embedding space after the text encoder.
> - No vocabulary projection is ever applied during optimization.
> - Text decoding happens **only once**, after optimization has converged, via our E2T model.
>
> This decoupling produces **a smooth and well-behaved optimization landscape**, eliminates projection noise, and dramatically improves efficiency and stability. Empirically, our embedding discrepancy is 4.4× smaller than token-space methods, and EDITOR converges reliably across datasets and architectures.
>
> Thus, the novelty of our method does not lie in using gradients per se, but in introducing a **new optimization formulation** where contextual-embedding optimization and natural-language decoding are **explicitly separated**. This conceptual shift resolves the central limitation of prior work and enables EDITOR to simultaneously achieve high reconstruction fidelity and human-readable prompt generation—capabilities that token-space gradient methods cannot attain.
>
>
>
> > ***[W2, Q1]: The notations and equations are incorrect. The cross-entropy loss and the MLE loss are not correctly defined in equation 4 and 6. Can the authors revisit and clarify the notations and equations. What is the difference between text decoder D and M.textdecoder in the algorithm. Also image is denoted by \mathbf{x} or at $x$ places.***
>
> We thank the reviewer for pointing out the notation issues.
>
>
> First, in equation (4) and (6), the presented formulas for $\mathcal{L}\_{\text{ML}}$ and $\mathcal{L}\_{\text{CE}}$ are mathematically sound and represent the standard **Negative Log-Likelihood (NLL)**, which is equivalent to **Cross-Entropy (CE) Loss** for sequence generation with discrete text outputs.
>
> To ensure clarity and consistency, **we have updated the manuscript to use consistent Maximum Likelihood notation for both objectives: $\mathcal{L}\_{\text{zero}}$ and $\mathcal{L}\_{\text{corr}}$**. Both minimize NLL.
>
>
> 1. Loss for the Zero-Step Model ($\mathcal{L}_{\text{zero}}$)
>
> $$\mathcal{L}\_{\text{zero}} = -\sum\_{(p,c)} \log P\_{\mathrm{zero}} \bigl(p \mid c\bigr).$$
>
> Here, $\sum_{(p,c)}$ runs over all training pairs, where $c$ is the input embedding and $p$ is the ground-truth prompt; the objective is standard NLL / MLE on $P\_{\mathrm{zero}}(p \mid c)$.
>
> 2. Loss for the Correction Model ($\mathcal{L}\_{\text{corr}}$)
>
> $$\mathcal{L}\_{\text{corr}} = -\sum\_{(p^{(k)},c)} \log P\_{\mathrm{corr}}(p^{(k)} \mid c).$$
>
> Here, $p^{(k)}$ is the ground-truth text that $M_{\mathrm{corr}}$ is trained to predict. In practice, $M_{\mathrm{corr}}$ is trained to satisfy $p^{(k)} = M_{\mathrm{corr}}\bigl(c,\; p^{(k-1)},\; \mathcal{T}(p^{(k-1)})\bigr)$, as defined in Eq.~(5). And $\mathcal{L}_{\text{corr}}$ is simply the NLL of $p^{(k)}$ under this model.
>
> Second, in Algorithm 1, $\mathcal{D}$ is intended to denote the image decoder of the diffusion model (i.e., the VAE decoder that maps latent representations back to pixel space), while $\mathcal{M}.\text{TextEncoder}$ refers to the text encoder of the same model. This was a typographical mistake in the original submission, and we thank the reviewer for pointing it out; in the revised version we explicitly rename $\mathcal{D}$ as “image decoder” for clarity.
>
> Finally, for the image notation, in Algorithm 1 we intentionally use the bold **symbol $x$** to emphasize the optimization target, whereas in the main text we use the standard $x$ to denote the target image. The two notations differ only in boldface for emphasis, and the underlying symbol is otherwise consistent. We do not use \mathbf{x} in the manuscript.

---

> ### Author Response · Authors · 2025-11-22
> **Reply to Reviewer Kp1H (2/2)**
>
> > ***[W3, Q2]: The approach does not consider recent multimodal architectures such as SD3. How does the approach extend to embedding spaces of the more recent architectures like SD3? Can this be mapped to the embedding of the captioning model?***
>
> We thank the reviewer for the question. We have already evaluated EDITOR on recent multi-encoder architectures, including Stable Diffusion 3.5 Medium in the original submission.
>
> Extending to SD3/SD3.5 only requires synchronously optimizing the latent embeddings of all text encoders (with the image reconstruction loss) and then applying the E2T model to the corresponding portion of the optimized embedding for each encoder. In our initial experiments, we follow this principle by training our E2T model on embeddings from the **CLIP-ViT/L** encoder, which is shared across these models. After optimization, we simply invert the CLIP-ViT/L component of the joint embedding using our trained E2T model to obtain the final prompt.
>
> The results are shown in **Table 6** :
>
> | Model                        | CLIP Score ↑ | LPIPS Score ↓ | Precision ↑ | Recall ↑ | F1 ↑  | PPL ↓   |
> |-----------------------------|--------------|----------------|-------------|----------|-------|---------|
> | Stable Diffusion v1.5       | 0.829        | 0.426          | 0.842       | 0.825    | 0.834 | 154.362 |
> | SDXL Turbo                  | 0.792        | 0.428          | 0.824       | 0.818    | 0.821 | 105.114 |
> | Stable Diffusion 3.5 Medium | 0.785        | 0.445          | 0.814       | 0.817    | 0.815 | 115.132 |
>
> hese results show that even when we train an E2T model for **only a single text encoder** in SDXL and SD3/3.5, our embedding-centric pipeline already performs strongly on advanced architectures with multiple encoders. This also indicates that EDITOR remains **effective and robust under modern Multimodal Diffusion Transformer (MMDiT)  architecture**.
>
>
> We additionally test the case where multiple SD3.5 text encoders each have their own E2T model. Concretely, we train separate E2T modules for CLIP-ViT/L and CLIP-ViT/G, jointly optimize the latent embeddings for both encoders, invert each optimized embedding with its corresponding E2T, and then select the better prompt. The results are:
>
> | Text Encoder              | CLIP Score ↑ | LPIPS Score ↓ | Precision ↑ | Recall ↑ | F1 ↑  | PPL ↓   |
> |---------------------------|--------------|----------------|-------------|----------|-------|---------|
> | CLIP-ViT/L                | 0.785        | 0.445          | 0.814       | 0.817    | 0.815 | 115.132 |
> | CLIP-ViT/L + CLIP-ViT/G   | **0.792**    | **0.441**      | 0.812       | 0.820    | 0.816 | 127.654 |
>
> Combining CLIP-ViT/L and CLIP-ViT/G improves CLIP and LPIPS over using CLIP-ViT/L alone, confirming that our embedding-level formulation and **E2T inversion naturally scale to multiple text encoders and can benefit from additional encoders by training the corresponding E2T modules**.
>
> Finally, we clarify that captioning models are **not** used as embedding sources in our pipeline. They only provide an initial hard prompt. Once a caption is generated, we always pass this text through the target diffusion model’s own text encoder to obtain the contextual embedding that EDITOR optimizes.
>
>
> > ***[W4, Q3]: Comparison to recent prompt inversion/search techniques such as [1]. How does it compare to the recent work on prompt search which also generates human readable prompts?***
>
> We thank the reviewer for suggesting STEPS as a relevant comparison. We performed a comprehensive empirical comparison with STEPS under the same query budget (1000). As shown below, our method significantly outperforms STEPS across all datasets and all evaluation metrics.
>
> |Dataset| Method | CLIP Score ↑ | LPIPS Score ↓ | Precision ↑ | Recall ↑ | F1 ↑  | PPL ↓   |
> | -------- | -------- | -------- | -------- | -------- | -------- | -------- |-------- |
> |MS COCO | STEPS |0.687 |0.480 |0.770 |0.847 |0.805 |5298.071 |
> | | EDITOR (Ours) |0.796| 0.414| 0.900| 0.920| 0.908| 80.659  |
> |LAION | STEPS |0.722 |0.476 |0.769 |0.796 |0.782 |7,040.633 |
> | | EDITOR (Ours) |0.826 |0.401 | 0.841 | 0.824 | 0.832 | 187.299  |
> |Flickr | STEPS |0.684 |0.466 |0.771 |0.851 |0.807 |6279.372 |
> | | EDITOR (Ours) |0.776  |0.424 |0.892 | 0.915 |0.900| 90.320  |
> |DiffusionDB | STEPS |0.749 |0.461 |0.777 | 0.807 |0.792 |7616.533 |
> | | EDITOR (Ours) |0.807 |0.385 |0.835 |0.813 |0.823 |105.185 |
>
> [1] STEPS: Sequential Probability Tensor Estimation for Text-to-Image Hard Prompt Search. CVPR 2025.

---

### Author Response · Authors · 2025-11-22
**Common Response**

We sincerely thank all reviewers for their careful reading of our manuscript and for providing constructive, insightful feedback. We are encouraged that the reviews highlight the strengths of our work, including the importance of interpretable prompt inversion, the effectiveness of our embedding-centric optimization pipeline, the quality and coherence of the recovered prompts, and the clarity of the overall presentation. We also greatly appreciate the reviewers’ detailed comments, which helped us substantially improve both the technical clarity and empirical completeness of the paper.

In response to the concerns raised across all reviews, we have conducted new experiments, added missing analyses, corrected notation and typographical issues, and expanded comparisons. In particular, the revised manuscript (with modifications highlighted in **blue**) includes:

- **New baseline**: STEPS (CVPR 2025), now fully included in all quantitative tables.
- **New dataset**: DiffusionDB, significantly broadening the evaluation distribution.
- **New ablations**: including *Zero-step vs. Correction Model*.
- **Statistical analysis**: confidence intervals, paired t-tests, and **per-image scatter plots** (Appendix F).
- **Clarified methodology**: corrected notation and equations, fixed typos, and clarified the roles of the image decoder and text encoder in Algorithm 1.

We hope that these changes address the reviewers’ concerns and further demonstrate the practicality, robustness, and broad applicability of EDITOR. We are grateful for the reviewers’ efforts and believe that the revised version is significantly strengthened thanks to their helpful and thoughtful suggestions.

---

### Author Response · Authors · 2025-12-01
**Rebuttal Summary**

We sincerely thank the reviewers for their thoughtful evaluations. We are encouraged by their assessment that:

- Our work tackles a significant challenge in reverse engineering diffusion models and effectively overcomes limitations of existing prompt inversion methods (Kp1H, zXzv).
- EDITOR introduces key innovations, including contextual-embedding optimization and E2T decoding within a clean, modular pipeline (SgN7, H81s).
- Our approach achieves strong empirical gains in image–prompt alignment, image similarity, and prompt fluency across datasets and models (Kp1H, H81s, SgN7, zXzv).
- The generated prompts are human-interpretable and valuable for provenance and downstream editing (H81s).
- Application studies highlight the method’s broader practical potential (zXzv).

We also thank the reviewers for their constructive criticisms, which helped us clarify our contributions and strengthen the paper. In our rebuttal, we specifically addressed the main concerns in a way that further reinforces the contributions highlighted above.

- **Clarifying and sharpening the novelty claim (Kp1H, H81s, zXzv).**
  We emphasized that prior token-space methods suffer from vocabulary projection, while existing continuous-embedding editing works use timestep-dependent, non-linguistic embeddings that cannot be decoded into a single prompt.
  We clarified that:
  - EDITOR’s **Initialize → Optimize → Decode** pipeline decouples embedding optimization from decoding, enabling stable and interpretable optimization.
  - PEZ and PH2P cannot benefit from initialization or stay on the prompt manifold due to projection, whereas EDITOR avoids projection entirely.
  - Editing methods like Prompt Tuning Inversion and NoiseCLR optimize embeddings unsuitable for prompt inversion, while EDITOR recovers a **global, decodable contextual embedding**.
    These points establish that EDITOR addresses the core limitations of both token-space and prior continuous-space approaches and forms a distinct paradigm.

- **Extending to modern architectures such as SD3 (Kp1H).**
  We clarified that SD3.5 results were already in the initial submission. The rebuttal further added experiments with multiple SD3.5 encoders and jointly optimized embeddings. Together with SD1.5 and SDXL-Turbo results, this shows EDITOR performs consistently well on advanced multi-encoder architectures and is **model-agnostic**.

- **Comparisons to recent prompt search and inversion methods (Kp1H).**
  We added a direct comparison with **STEPS**, showing that under the same query budget EDITOR outperforms STEPS across all datasets and metrics, reinforcing our empirical contribution.

- **Improving robustness and evaluation scale with new datasets (H81s, SgN7).**
  To address dataset scale concerns, we added results on **DiffusionDB**, whose user-generated prompts and images follow a distribution entirely different from COCO, LAION, and Flickr.
  Even under this significant shift, EDITOR outperforms PEZ, PH2P, and VGD across all metrics. Confidence intervals and paired tests confirm the improvements are statistically significant and stable.

- **Fixing notation and equation issues (Kp1H, zXzv).**
  We corrected notation errors, clarified key equations, and fixed typos and ambiguous descriptions for improved clarity.

- **Addressing complexity and additional ablations (H81s, zXzv, SgN7).**
  We clarified that E2T is trained once offline with small inference overhead. We also included ablations on the zero-step and correction models, seed and initializer sensitivity, and noted that beam-search ablations were already in the initial submission.

- **Editability and semantic stability (SgN7).**
  We explained that EDITOR avoids projection drift and maintains strong reconstruction and text-alignment metrics, supported by downstream editing results.

**In summary, the rebuttal directly addresses all critical concerns, especially regarding novelty, baselines, and evaluation scale**. After Reviewer zXzv provided the missing references, we further clarified why prior continuous-embedding editing methods cannot be applied to prompt inversion, strengthening our argument about novelty. The rebuttal reinforces the core contributions recognized by the reviewers:
1. A new and modular **initialize–optimize–decode** pipeline that performs embedding-space optimization and natural-language decoding in a decoupled manner, providing a **novel, clean, and effective formulation for prompt inversion**.
2. **Overcoming the core limitations of prior prompt inversion and image-editing inversion approaches**, including vocabulary. projection, discrete optimization instability, and timestep-dependent non-linguistic embeddings that cannot be decoded into a single prompt.
3. **Strong and statistically validated improvements across architectures, baselines, and datasets**.

---

### Meta-Review · Area_Chair_2jLf · 2026-01-07

**Summary:**

The submission presents an approach for prompt inversion from generated images, but the reviewers expressed concerns about its limited novelty and insufficient evaluation. The paper mainly combines existing methods (Kp1H, zXzv), and the validation using only 100 images is not convincing (H81s, SgN7).

**Reviewer Concerns:**

While the authors provided additional clarifications and experiments in their rebuttal, addressing aspects of clarity, technical accuracy, and methodological robustness, the key issues—such as the novelty and performance on multiple full datasets—remain inadequately addressed.

**Reviewer Scores:**

Reviewers Kp1H and zXzv questioned the innovation of the work. Although the authors emphasized that the novelty lies in the overall pipeline design, the pipeline primarily enables the embeddings to better align with the target image, which lacks substantial innovation. Therefore, the reviewers are likely to maintain their original scores.

Reviewers H81s and SgN7 expected the authors to provide evaluations on full datasets. However, the authors did not directly address this request, and the reviewers may maintain or even lower their scores as a result.

---

### Decision · Program_Chairs · 2026-01-26

Reject